# Coordination of rapid cholinergic and dopaminergic signaling in striatum during spontaneous movement

Mark Howe[†]*, Imane Ridouh, Anna Letizia Allegra Mascaro[‡], Alyssa Larios, Maite Azcorra, Daniel A Dombeck*

Department of Neurobiology, Northwestern University, Evanston, United States

**Abstract** Interplay between dopaminergic and cholinergic neuromodulation in the striatum is crucial for movement control, with prominent models proposing pro-kinetic and anti-kinetic effects of dopamine and acetylcholine release, respectively. However, the natural, movement-related signals of striatum cholinergic neurons and their relationship to simultaneous variations in dopamine signaling are unknown. Here, functional optical recordings in mice were used to establish rapid cholinergic signals in dorsal striatum during spontaneous movements. Bursts across the cholinergic population occurred at transitions between movement states and were marked by widespread network synchronization which diminished during sustained locomotion. Simultaneous cholinergic and dopaminergic recordings revealed distinct but coordinated sub-second signals, suggesting a new model where cholinergic population synchrony signals rapid changes in movement states while dopamine signals the drive to enact or sustain those states.

DOI: https://doi.org/10.7554/eLife.44903.001

**\*For correspondence:**
mwhowe@bu.edu (MH);
d-dombeck@northwestern.edu
(DAD)

**Present address:** [†]Department of Psychological and Brain Sciences, Boston University, Boston, United States; [‡]European Laboratory for Non-linear Spectroscopy, Florence, Italy and Neuroscience Institute, National Research Council, Pisa, Italy

**Competing interests:** The authors declare that no competing interests exist.

## Introduction

Striatum cholinergic interneurons (ChIs) have long been recognized as critical for normal functioning of the basal ganglia in the control of movement and in learning and responding to motivational and salient stimuli (*Apicella et al., 1996*; *Apicella et al., 1997*; *Apicella et al., 1991*; *Collins et al., 2016*; *Kimura et al., 1984*; *Kondabolu et al., 2016*; *Schulz et al., 2011*; *Kaneko et al., 2000*). Although they comprise a small proportion of the total cell population (~1–3%) in striatum (*Kimura et al., 1980*), they are believed to contribute to debilitating motor and cognitive symptoms in a range of disorders including Parkinson's Disease (PD) (*McGeer et al., 1961*; *Barbeau, 1962*; *Tanimura et al., 2018*; *Bordia et al., 2016a*; *Bordia et al., 2016b*). Anti-cholinergic (muscarinic) drugs are recognized as the earliest treatment capable of alleviating some of the debilitating movement deficits characteristic of PD (*McGeer et al., 1961*; *Barbeau, 1962*; *Tanimura et al., 2018*). This clinical observation led to the hypothesis that loss of dopaminergic (DA) innervation to the dorsal striatum from the substantia nigra pars compacta (SNc) in PD disrupts a dynamic balance between acetylcholine and dopamine signaling (*Barbeau, 1962*). Targeted manipulations of dorsal striatal ChIs in rodents, either pharmacologically or optogenetically, produce changes in behavior roughly consistent with this hypothesis (*Kondabolu et al., 2016*; *Kaneko et al., 2000*; *Bordia et al., 2016b*). Unilateral ablation of ChIs results in ipsilateral turning behavior opposite to the effects of unilateral DA terminal lesions (*Kaneko et al., 2000*). Bilateral optogenetic stimulation of dorsal striatum ChIs reduces gross spontaneous locomotion (*Kondabolu et al., 2016*) (an effect counter to stimulation of DA [*Howe and Dombeck, 2016*; *da Silva et al., 2018*]), while optogenetic inhibition can alleviate motor deficits of L-DOPA induced dyskinesias in rodent PD models (*Bordia et al., 2016b*; *Maurice et al., 2015*; *Ztaou et al., 2016*). These findings suggest that cholinergic signaling opposes the pro-kinetic actions of dopamine in the striatum. However, while naturally occurring

signaling patterns of DA input to the striatum during movement have been described (*Howe and Dombeck, 2016*; *Patriarchi et al., 2018*), signaling in cholinergic interneuron networks, and the precise coordination between ChI and DA signaling occurring during movements, remain unclear.

In behaving rodents and primates, extracellular single-unit recordings of tonically active, putative ChIs (TANs) have revealed rapid (<200 ms latency), synchronized responses to salient, neutral and positive and negative conditioned stimuli and to primary rewards and punishments (*Apicella et al., 1996*; *Apicella et al., 1991*; *Kimura et al., 1984*; *Aosaki et al., 1994a*; *Ravel et al., 1999*; *Ravel et al., 2003*; *Morris et al., 2004*; *Apicella, 2017*; *Blazquez et al., 2002*), but rarely to spontaneous movements. Stimulus evoked TAN responses often consist of an initial pause in firing rate (occasionally preceded by a short burst of firing) followed by a burst in firing rate. The magnitude and duration of the pause/burst is modulated by learning and depend in some cases on the nigrostriatal dopamine system (*Ravel et al., 2003*; *Aosaki et al., 1994b*; *Aosaki et al., 1995*). These findings indicate that striatum ChI signaling may play a role, along with DA, in modulating long term plasticity for goal directed learning (*Aosaki et al., 1994b*). Based on recordings and manipulations, TAN bursts and pauses to conditioned and salient sensory stimuli have also been proposed to drive rapid attentional orienting, goal-directed action, and set-shifting (*Collins et al., 2016*; *Ding et al., 2010*; *Aoki et al., 2018*; *Aoki et al., 2015*; *Atallah et al., 2014*; *Tzavos et al., 2004*). Experimental evidence directly linking cholinergic interneuron signaling to rapid changes in movement has been less common. Two studies in freely moving rodents have described heterogeneous positive and negative responses in dorsal and ventral striatum TANs aligned to movements during a reward-directed instrumental task (*Yarom and Cohen, 2011*; *Benhamou et al., 2014*). In primates, TANs do not generally respond to simple cued arm movements, but one study described small responses to self-timed movements (*Lee et al., 2006*) and another showed a relationship between cue evoked TAN responses and the probability to respond in eye blink conditioning (*Blazquez et al., 2002*). Another described TAN modulation that varied with the force required for cued arm movements (*Nougaret and Ravel, 2015*). However, it remains unclear whether these movement-related TAN responses reflect instrumental task contingencies, cue responses or changes in movement per se. Moreover, in-vivo extracellular unit studies have been limited in their ability to discriminate true ChI responses from other striatal cell types (*Atallah et al., 2014*; *Gage et al., 2010*). Thus, it remains unclear how or whether ChIs participate in spontaneous movement control independently of discrete action-outcome contingencies, outcome predicting cues, or salient external stimuli.

The effects of ChI signaling on striatal microcircuits and dopaminergic afferents have been investigated intensively in vitro and through manipulations in vivo. From these studies, a complex picture has emerged in which ChIs are capable of bi-directionally influencing cell-type specific excitability of both medium spiny projection neurons (SPNs) and striatal interneurons (*Goldberg et al., 2012*; *Lim et al., 2014*; *Oldenburg and Ding, 2011*; *Higley et al., 2009*) and can directly alter the release probability from dopaminergic terminals on relatively rapid timescales (10 s-100s of ms) (*Di Chiara et al., 1994*; *Threlfell et al., 2012*). Ach release can also modulate long term pre and post-synaptic plasticity (minutes to days) on SPNs (*Wang et al., 2006*; *Lee et al., 2016*). In turn, SPNs, interneurons, and DA afferents, along with excitatory glutamatergic inputs from thalamus and cortex provide powerful and rapid modulation of the cholinergic interneurons (*Aosaki et al., 1994b*; *Ding et al., 2010*; *Di Chiara et al., 1994*; *Matsumoto et al., 2001*; *Raz et al., 1996*; *Straub et al., 2014*; *Witten et al., 2010*; *Zhang et al., 2018*; *Zucca et al., 2018*). These extensive input/output connections provide enormous possibilities for ChIs to exert positive and negative feedback on striatal circuits. Determining the circuit dynamics of ChIs resulting from all of these mechanisms and the relation of the dynamics to movement control and learning will require large-scale measurements of cholinergic interneuron networks and other circuit elements, including DA inputs, which coordinate closely with ChI signaling.

Electrophysiological measurements of naturally occurring ChI signaling in behaving animals have to date been limited to only a few (typically 2–4) putative ChIs recorded simultaneously and are challenged by difficulties in reliable cell identification. Here, we address this limitation by utilizing 2-photon calcium imaging and fiber photometry in conjunction with genetic cell type specific labeling to measure signaling across the ChI population and from up to 15 identified single ChIs simultaneously in the dorsal striatum in head-fixed, mobile mice. We describe previously unknown rapid, transient signals across the ChI population and in single neurons locked to onsets of spontaneous locomotion, short jerky movements, and to locomotor 'resets' in the absence of salient stimuli or reward

prediction. We show that movement initiations are marked by highly synchronous transients across the cholinergic interneuron network, but that synchrony (but not single-neuron activity) diminishes significantly as mice transition to continuous locomotion. We then establish a new approach to measure *simultaneous* dynamics in both ChIs and dopaminergic projection axons to the dorsal striatum, which demonstrates precise sub-second coordination of dopaminergic and cholinergic signals in relation to spontaneous movement. This coordination varies dynamically as a function of the animals' behavioral state, suggesting that dopamine and acetylcholine may act both synergistically and competitively to regulate different aspects of movement. From these results, we propose a new model where cholinergic population synchrony signals rapid changes in movement states while dopamine signaling provides rapid drive to enact and sustain new movement states. These findings provide insight into how spontaneous movement is regulated by neuromodulator dynamics in the striatum and suggest new possibilities for how DA/Ach interactions occur at the circuit level during behavior.

## Results

To determine whether and how populations of dorsal striatum ChIs signal spontaneous changes in movement, independently of task contingencies, we first performed acute fiber photometry in Chat-cre mice (n = 20 mice, 1–3 sessions/mouse) expressing GCaMP6f (*Chen et al., 2013*) specifically in cholinergic interneurons (*Figure 1A,B*, *Figure 1—figure supplement 1A–D*). Mice were head-fixed with their limbs resting on a cylindrical treadmill and spontaneously transitioned between periods of movement and rest (*Figure 1A–B*). Very little change in ChI population signaling was observed when mice were at rest but positive going transients were consistently elevated during spontaneous movement (*Figure 1B–C*; mean DF/F, p<1e-19 Wilcoxon rank-sum test). ChI population signals increased rapidly and reliably approximately 100 ms prior to spontaneous movement onsets from rest (*Figure 1E–F*). Very small 'micro' movements occurring at velocities and accelerations below our rest threshold did not elicit significant ChI responses (*Figure 1—figure supplement 1E*). Broadly, movement onsets from rest could be classified into two types: transitions to continuous locomotion (*Figure 1B,G,K*) and short duration 'jerks', consisting of rapidly terminated forward and backward movements (*Figure 1B,H,L*). Significant elevations in ChI signaling were observed for both of these movement onset types with similar timing and amplitudes (*Figure 1G,H*). A gradual decrease in ChI signal to baseline was observed at locomotion terminations (*Figure 1B,I,M*). During continuous, high velocity locomotion periods, significantly less transient signaling was present relative to movement onset periods (*Figure 1B,D*; p<0.01 Wilcoxon rank-sum test), though transient signaling during these periods was greater than during rest (*Figure 1B,D*; p<3e-13, Wilcoxon rank-sum test). The largest ChI transients (top quartile) occurring post-onset were associated with large accelerations at low velocity, corresponding to movement resets in ongoing locomotion (*Figure 1B,J,N*, *Figure 1—figure supplement 1F*) where the mouse rapidly transitioned to near rest then re-accelerated to resume locomotion. Less signal change was associated with accelerations occurring during high velocity, continuous locomotion (*Figure 1—figure supplement 1F*). ChI movement related signals were observed in the absence of task contingencies, changes in salient sensory stimuli, and before any exposure to rewards (*Figure 1—figure supplement 1G–H*). These results demonstrate that transient dorsal striatum ChI population signals reflect rapid transitions in internally generated spontaneous movement, from rest to movement or during resets in locomotion, independently of changes in reward prediction or experimentally generated salient stimuli.

Previous studies have shown that dopaminergic signals to spontaneous movements and to unpredicted rewards observe a dorsal-ventral gradient in the striatum (*Howe and Dombeck, 2016*; *Parker et al., 2016*). We therefore tested whether similar signaling gradients were present for the ChI population. GCaMP6f expression was confirmed in ChAT+ cells of the ventral striatum (NAc core) of ChAT-cre mice (*Figure 1—figure supplement 1A–D*). Robust transients were observed in both dorsal and ventral recordings from the same mice (*Figure 2A*), but only dorsal ChI signals rapidly increased at spontaneous movement onsets from rest (*Figure 2B*). Moreover, unlike dorsal striatum, ventral striatum transients did not exhibit consistent timing with respect to spontaneous mouse accelerations (*Figure 2C*). These data indicate that, similar to dopamine projections, spontaneous movement signals in ChIs are different across the dorsal-ventral axis of the striatum.

Putative ChIs (TANs) recorded with in-vivo electrophysiology respond with rapid pauses and bursts to unpredicted rewards and reward-predictive stimuli (*Apicella et al., 1996*; *Apicella et al.,*

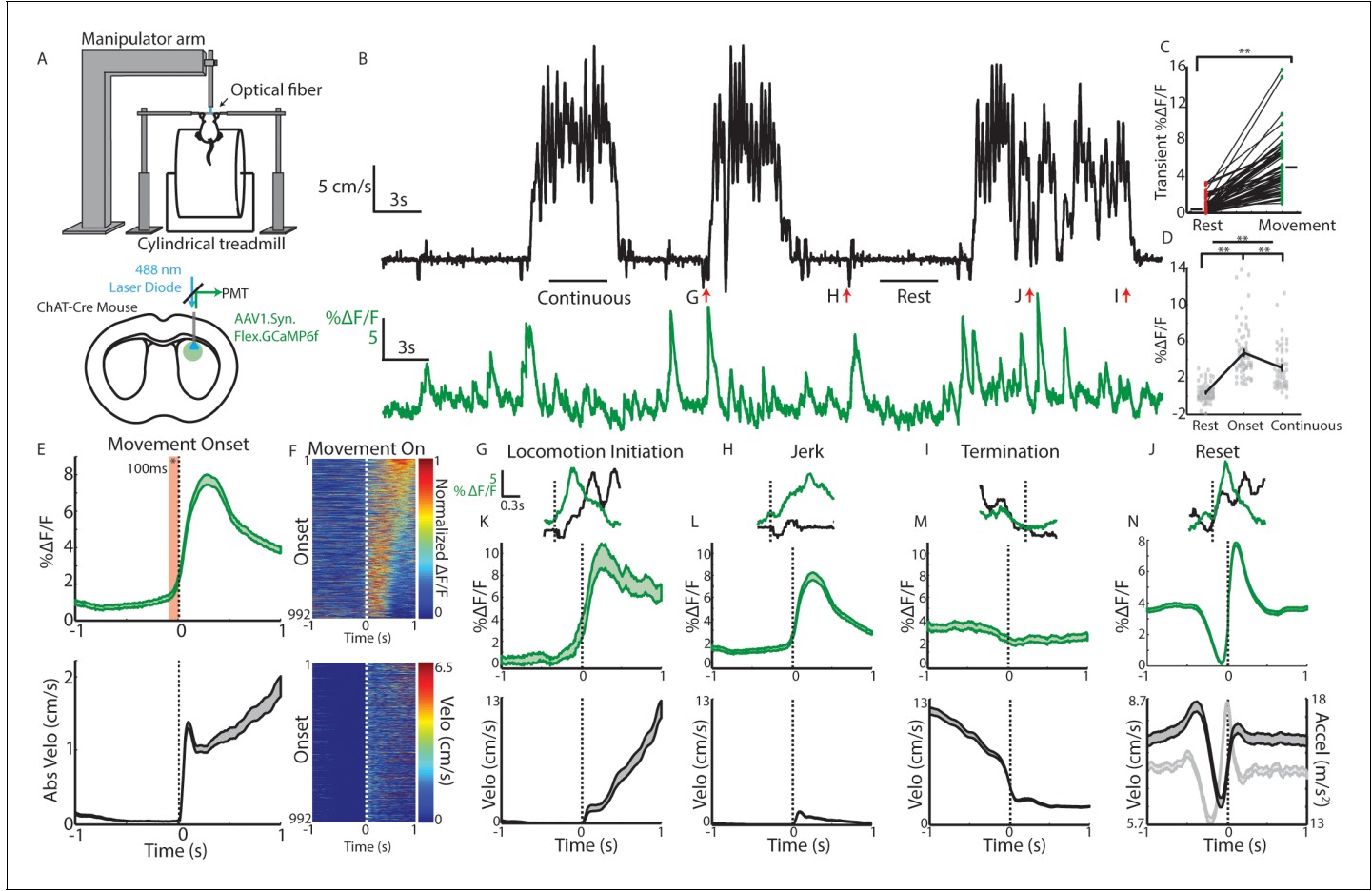

**Figure 1.** Cholinergic interneuron populations are rapidly activated at spontaneous movement onsets. (A) Schematic of experimental setup. Acute fiber photometry recordings from GCaMP6f expressing ChIs in dorsal striatum (bottom) in head fixed mice moving spontaneously on 1D treadmill in darkness (top). (B) Representative population fluorescence changes (DF/F, green) from ChIs during spontaneous treadmill movement (velocity, black). (C) Mean transient DF/F for each recording (n = 62 sessions, 19 mice) during rest or movement periods. (D) Mean DF/F during all rest, onset, and continuous locomotion periods for the sessions in (C). (E) Mean DF/F (top) and velocity (bottom) aligned on the onset of all clean movement onsets from rest (see Materials and methods, n = 992 onsets, 16 mice). Shaded region DF/F greater than mean of all rest periods, p<0.01 Wilcoxon Rank-Sum test. (F) All peak normalized traces (top) and velocites (bottom) aligned on movement onsets and sorted by peak responses. (G) Zoomed DF/F (green) and velocity (black) for the locomotion initiation period indicated in (B). (H) Zoomed jerk period indicated in (B). (I) Zoomed termination period indicated in (B). (J) Zoomed behavior reset indicated in (B). (K) Mean DF/F (top) and velocity (bottom) aligned on the onset of all clean locomotion initiations from rest (n = 83 onsets, nine mice). (L) Mean DF/F (top) and velocity (bottom) aligned on the onset of all clean jerks from rest (n = 543 jerks, 18 mice). (M) Mean DF/F (top) and velocity (bottom) aligned on the onset of all locomotion terminations (n = 251 terminations, 19 mice). (N) Mean DF/F (top) and velocity and acceleration (bottom) aligned on the onset of all positively-going transients during motion (n = 4192 transients, 19 mice). **p<1×10–6 Wilcoxon Rank Sum Test. Shaded regions represent ±SEM.

DOI: https://doi.org/10.7554/eLife.44903.002

The following figure supplement is available for figure 1:

**Figure supplement 1.** Fiber photometry recording sites, histology, and additional properties of movement related ChI signaling.

DOI: https://doi.org/10.7554/eLife.44903.003

1997; *Apicella et al., 1991*; *Kimura et al., 1984*; *Aosaki et al., 1994a*; *Ravel et al., 1999*; *Apicella, 2017*; *Aosaki et al., 1994b*; *Apicella, 2007*; *Apicella et al., 2009*); we tested whether similar signals were present across populations of ChIs in the dorsal and ventral striatum. Delivery of unpredicted water rewards through a spout near the animals' mouth elicited responses in dorsal striatum ChIs, that on average, consisted of a short latency increase followed by a small decrease aligned to the audible solenoid click associated with water valve opening (*Figure 2D*). While reminiscent of the previously reported burst/pause observations, this response was variable and both the increase and decrease were not reliably present across individual trials (*Figure 2E*). In contrast, ventral striatum

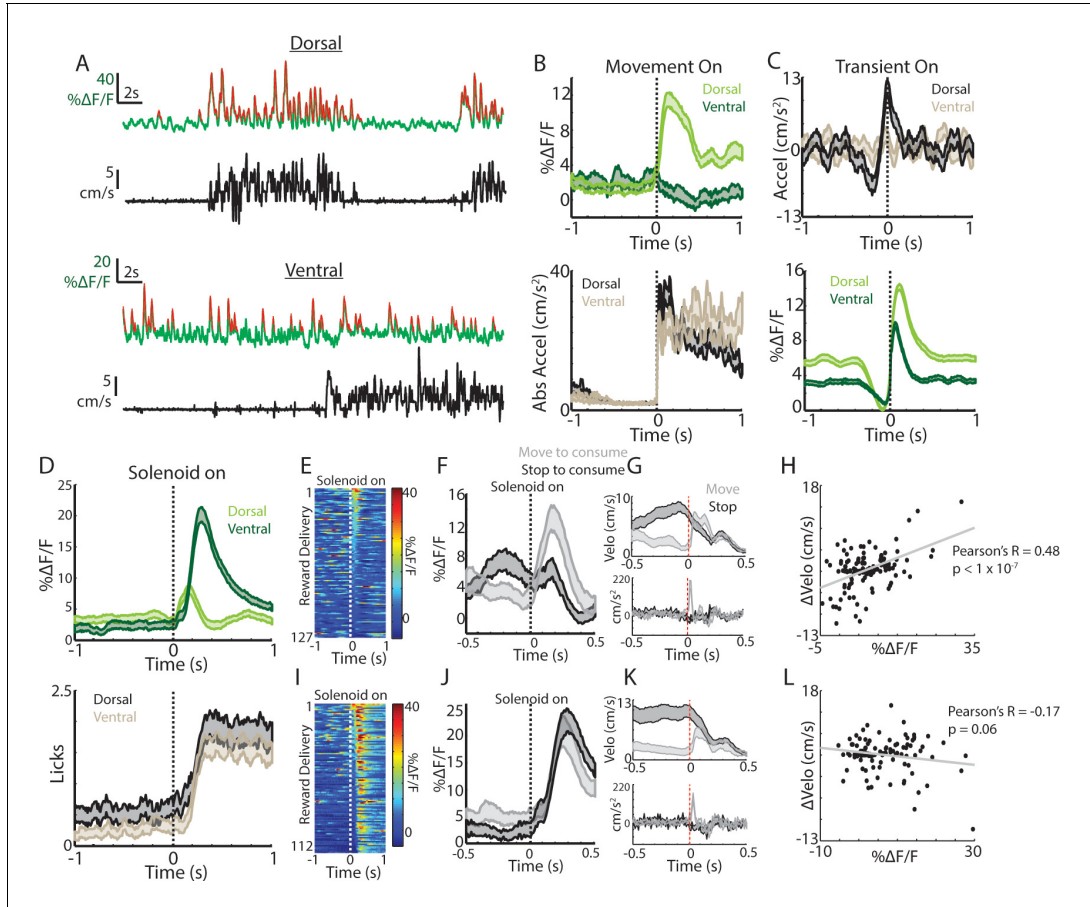

**Figure 2.** Movement and reward related signals in ChI populations differ across the dorsal-ventral axis of the striatum. (A) Representative population fluorescence changes (DF/F, green) and velocity (black) from dorsal (top) and ventral (bottom) striatum ChIs during spontaneous movement. Red indicates identified significant positive-going transients. (B) Mean population DF/F (top) and acceleration (bottom) aligned on movement initiations from rest across all dorsal (n = 118 onsets, three mice) and ventral (n = 66 onsets, three mice) ChI recordings without reward deliveries. (C) Mean acceleration (top) and transient population DF/F (bottom) aligned on onsets of significant positive-going transients across dorsal (n = 1215 transients) and ventral (n = 851) ChI recordings without reward. (D) Mean DF/F and spout licking aligned on the unpredicted triggering of the solenoid reward valve for dorsal (n = 127 rewards, 10 mice) and ventral (n = 112 rewards, eight mice) ChI population recordings. (E) DF/F for all unpredicted reward deliveries aligned on solenoid valve trigger for dorsal ChI recordings sorted by peak response. (F) Mean dorsal ChI DF/F for reward deliveries where the mouse stopped to consume the reward from locomotion (bottom quartile of velocity changes at reward delivery, n = 32) or accelerated from a resting state (top quartile, n = 32). (G) Mean velocity (top) and acceleration (bottom) for the deliveries plotted in F. (H) Change in velocity vs the mean dorsal ChI DF/F for each reward delivery. (I) Same as E for ventral ChI recordings. (J) Same as F for ventral ChI recordings (n = 28). (K) Same as G for ventral recordings. (L) Same as H for ventral recordings. Shaded regions represent ±SEM.

DOI: https://doi.org/10.7554/eLife.44903.004

The following figure supplement is available for figure 2:

**Figure supplement 1.** Dorsal and ventral ChI populations respond differently to unpredicted reward consumption and to conditioned cues.

DOI: https://doi.org/10.7554/eLife.44903.005

ChIs exhibited robust and consistent increases on nearly every trial following reward delivery (*Figure 2D,I*).

Mice displayed changes in movement at the reward delivery time that varied depending upon the movement state (running or resting) prior to consumption, we therefore examined whether the timing and direction of movement changes could account for the variability in dorsal and ventral striatum ChI responses (*Figure 2F–H,J–L*, *Figure 2—figure supplement 1B*). When reward was delivered at rest, mice briefly moved their bodies to initiate consumption from the water spout (brief acceleration; *Figure 2G,K*). When rewards were delivered during movement, mice rapidly decelerated to initiate consumption (*Figure 2G,K*). The dorsal striatum ChI response was significantly

positively correlated with these changes in velocity at reward delivery (*Figure 2F–H*, R = 0.48, p<1.7e-8) but the response in ventral striatum was not (*Figure 2J–L*, R = −0.17, p=0.06). Consistent with this result, when responses were re-aligned to the time at which consumption began (first detected spout lick after the change in movement), the dorsal striatum ChI response diminished, while the ventral ChI response remained (*Figure 2—figure supplement 1B*). In fact, the ventral ChI response increase on average preceded the first lick, indicating a contribution of reward prediction associated with the solenoid click (*Figure 2—figure supplement 1B*). These data indicate that dorsal and ventral ChIs exhibit different responses to unpredicted rewards and that changes in movement at reward consumption could account for much of the dorsal striatum ChI population response. This differential expression of spontaneous movement and unpredicted reward responses across the dorsal-ventral axis parallels that reported for striatum dopamine signaling (*Howe and Dombeck, 2016*; *Parker et al., 2016*).

Notably, we did observe a prominent triphasic increase/decrease/increase signal in dorsal striatum ChIs to a conditioned visual stimulus paired with reward similar to those described in recordings of TANs with electrophysiology (*Aosaki et al., 1994a*) (*Figure 2—figure supplement 1C,G*). This response could not be fully explained by changes in movement (*Figure 2—figure supplement 1D, G*), indicating that dorsal striatum ChIs do display a response to conditioned cues that is at least partly independent of movement changes. Ventral ChIs, in contrast, exhibited a sustained increase at the CS onset and a sharp increase at the delivery of the predicted reward (*Figure 2—figure supplement 1E–G*) consistent with other rodent ventral TAN recordings (*Atallah et al., 2014*). Moreover, ventral ChIs exhibited a small increase after movement onsets for sessions after experience with unpredicted rewards (but not before, *Figure 2—figure supplement 1A*), indicating that ventral ChIs are not sensitive to spontaneous movement per se but may be influenced by changes in reward prediction that coincide with changes in movement, a property also present in ventral striatum dopamine signaling (*Ko and Wanat, 2016*; *Syed et al., 2016*) and midbrain cell body firing (*Coddington and Dudman, 2018*). These findings demonstrate that ChI signals differ strikingly in their responses to cues, rewards, and to movement across the dorsal-ventral axis of the striatum, similarly to dorsal-ventral gradients in DA signaling.

Fiber photometry measures an aggregate weighted average from active cells (cell bodies, axons and dendrites) within a small region (~100–200 μm radius sphere) around the optic fiber (*Cui et al., 2013*), but single neurons may coordinate in multiple ways to generate the observed population transients (*London et al., 2018*; *Rehani et al., 2019*). To investigate how networks of single cholinergic interneurons represent changes in movement, we performed 2-photon imaging of GCaMP6f expressing ChIs at subcellular resolution in dorsal striatum (n = 6 mice, Methods, *Figure 3* and *Figure 3—figure supplement 1*). The size of our imaging fields ranged from 100 μm to 700 μm, enabling us to capture signals from up to 15 ChI cell bodies simultaneously, along with many of their dendrites and axons. Since ChIs fire tonically at ~3–10 Hz (*Wilson et al., 1990*), baseline fluorescence in cell bodies presumably reflected ongoing tonic firing, while large $Ca^{2+}$ transients presumably represented bursts of action potentials (*Chen et al., 2013*). These transients were highly synchronized and reliably present across proximal and distal dendritic branches of the ChIs, indicative of regenerative events (*Figure 3C–D*, backpropagating action potentials or dendritic spikes [*Rehani et al., 2019*; *Spruston et al., 1995*]).

Consistent with photometry recordings, nearly all single ChIs exhibited significantly larger and more frequent calcium transients during all movement periods relative to rest (*Figure 3E*, p=4.1e-87 Wilcoxon rank sum test across all cells, 266/268 cells p<0.01). Single ChIs exhibited similar rapid increases to population photometry measures at general movement onsets (*Figure 3G–H*) and increases at jerks and locomotion initiations (*Figure 3I–J*). Unlike the population photometry measures, however, transient signaling in single ChIs was also prominent during continuous locomotion to a similar extent to locomotion onsets (*Figure 3F,G*). Imaging the same fields with 800 nm excitation light (rendering GCaMP fluorescence static, that is non-activity dependent) or in separate mice expressing the static red indicator td-tomato did not reveal significant positive going transients or rapid increases at movement onsets (*Figure 3—figure supplement 1*), indicating that movement induced artifacts in the fluorescence measurements could not account for the movement-related transients in single ChIs. In summary, transient encoding of spontaneous movement is present at movement onsets and during continuous locomotion in the majority of dorsal striatum ChIs.

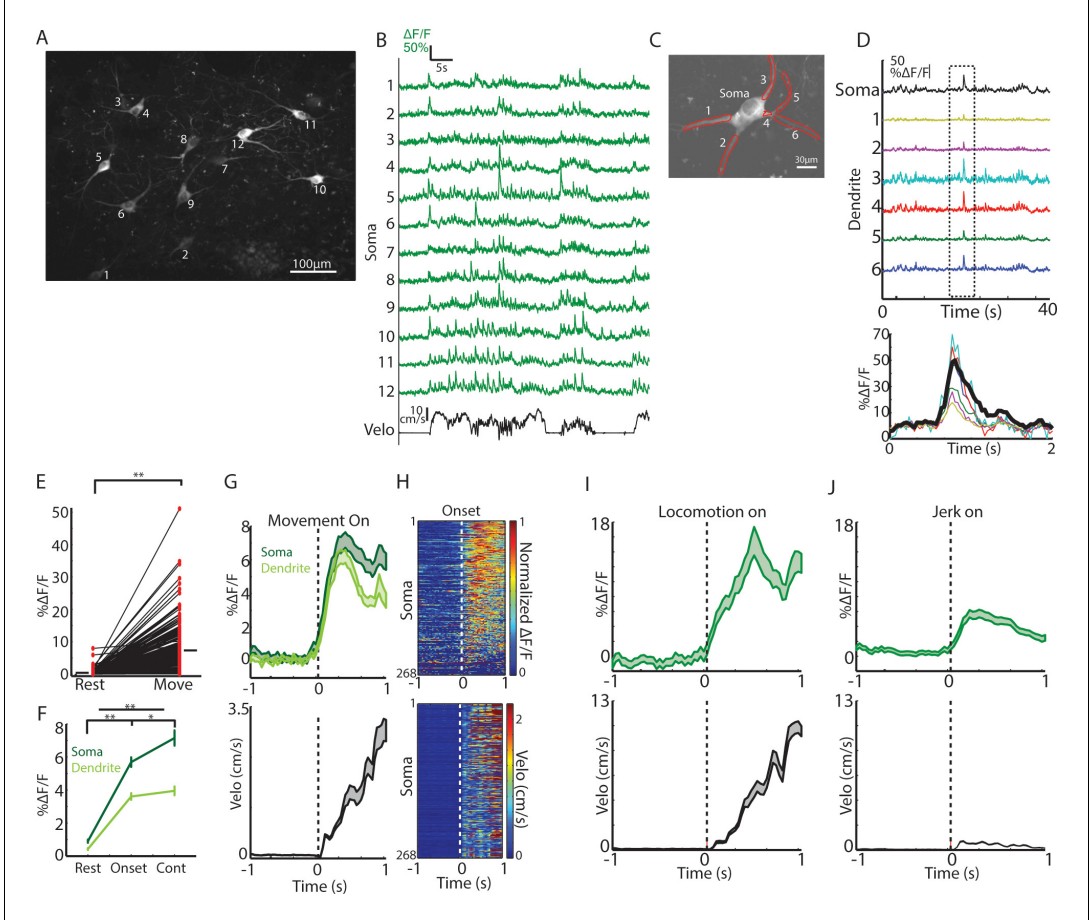

**Figure 3.** Single ChIs signals are rapidly elevated at movement onsets and throughout locomotion. (A) Mean fluorescence projection image from 2-photon imaging of a representative field containing multiple distinct GCaMP6f expressing ChIs. (B) DF/F for each of the ROIs in A during a representative behavior period (velocity, black). (C) Mean fluorescence projection image of a single dorsal striatum ChI. Red ROIs indicate dendritic branches. (D) DF/F from the ROIs in C. Note the presence of synchronized transients across all dendrites (period in dashed box zoomed at bottom). (E) Mean transient DF/F during rest and movement periods for all ChIs somata (n = 268 neurons, six mice). (F) Mean transient DF/F across ChIs somata and dendrites for rest, movement onset, and continuous locomotion periods. (G) Mean DF/F (top) across all ChIs and velocity (bottom) aligned on clean movement onsets from rest. (H) Peak normalized mean DF/F for each ChI soma (top) and velocity (bottom) sorted by peak responses at movement initiation. (I) Mean DF/F (top) across ChI somata (n = 96 cells, six mice) and velocity (bottom) aligned on clean locomotion initiations from rest. (J) Mean DF/F (top) across ChI somata (n = 213 cells, six mice) and velocity (bottom) aligned on clean jerk onsets from rest. *p<0.01, **p<1e-8, Wilcoxon rank sum test. Shaded regions represent ±SEM.

DOI: https://doi.org/10.7554/eLife.44903.006

The following figure supplement is available for figure 3:

**Figure supplement 1.** Movement related ChI signals are not generated by motion artifact.

DOI: https://doi.org/10.7554/eLife.44903.007

The observation that mean transient changes in dorsal striatum ChI population fluorescence as measured with photometry (*Figure 1D*), but not single cell transients (*Figure 3F*), decreased from locomotion initiations to continuous locomotion, raised the possibility that transient *synchrony* across the population of ChIs, rather than signaling *amplitude* across single cells in the population, may be modulated during behavior. Synchrony of firing across the ChI network has been identified in vitro as an important determinant of its influence on pre-synaptic dopamine release from axon terminals (*Threlfell et al., 2012*). High correlations across pairs or triplets of TANs have been reported to conditioned stimuli and rewards in behaving primates (*Morris et al., 2004*; *Raz et al., 1996*), but synchrony has never been evaluated across larger networks of ChIs (>10) during spontaneous movement. Over all behavior periods, activity across ChIs was highly correlated, even for pairs of neurons separated by >400 um (*Figure 4A–E*). Transients were synchronized with a peak at zero

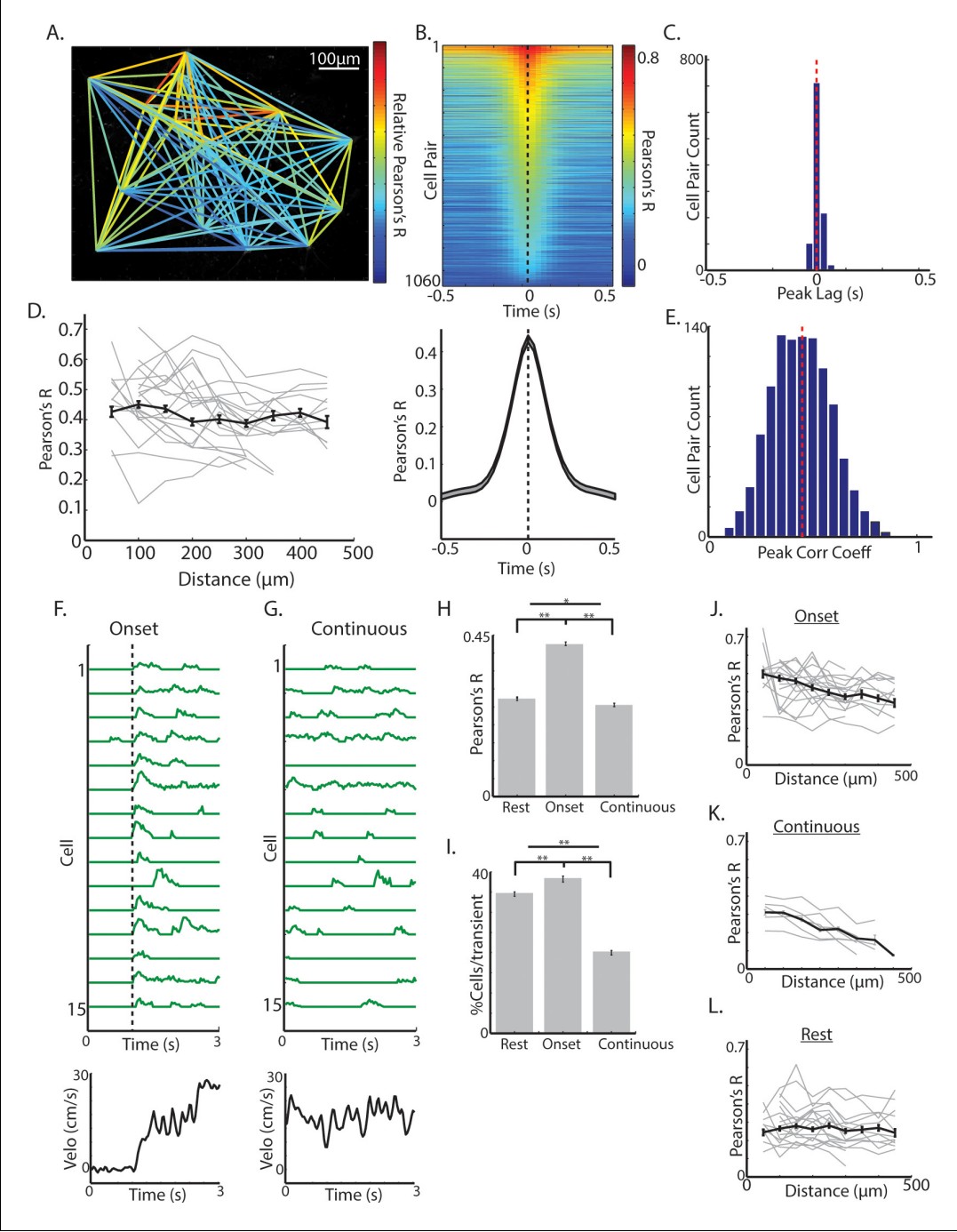

**Figure 4.** Synchrony across ChIs varies with spontaneous movement state. (**A**) Mean fluorescence projection image from 2-photon imaging with color of lines connecting each cell pair representing relative DF/F correlations of those cells across time. (**B**) All cell pair cross-correlations sorted by peak correlation (top) and the mean across all cell pairs (n = 1060 pairs, six mice; bottom). (**C**) Histogram of peak lags across cell-pairs. Red dashed line is the mean lag. (**D**) Mean correlation vs distance for all periods and all cell pairs (black line) and for cell pairs in individual fields containing >10 pairs (each line is a single field). (**E**) Histogram of peak correlation coefficients across all cell pairs. Red line is the mean. (**F**) Transient DF/F from 15 cells aligned on a representative locomotion initiation (velocity, bottom). (**G**) Same cells as F during a period of continuous locomotion. (**H**) Mean correlation coefficient for cell pairs at rest (n = 852 pairs), at movement onsets (n = 842 pairs), and during continuous locomotion (n = 524 pairs). (**I**) Mean percent of cells active for a given transient event during rest (n = 1816 transients), onsets (n = 1158), and continuous locomotion (n = 1059). (**J**) Mean correlation vs distance for all cell

*Figure 4 continued on next page*

*Figure 4 continued*

pairs during movement onsets (grey lines are each individual field with >10 pairs). (K) Same as J for continuous locomotion. (L) Same as J,K for rest periods. Shaded regions and error bars represent ±SEM. **p<1×10⁻⁷, *p<0.05; Wilcoxon Rank-Sum test.

DOI: https://doi.org/10.7554/eLife.44903.008

The following figure supplement is available for figure 4:

**Figure supplement 1.** Network synchrony is similarly elevated at locomotion and jerk onsets.

DOI: https://doi.org/10.7554/eLife.44903.009

phase lag, and this timing was consistent across cell pairs (*Figure 4B–C*). Also across all behavior periods, there was a weak but significant negative relationship between the magnitude of pairwise correlations and distance between cell pairs separated by distances < 500 um (R = −0.08, p=0.005, *Figure 4D*). This data indicates that ChI networks are highly synchronized during behavior across large regions of dorsal striatum and positioned to potentially influence dopamine release from local axon terminals.

We then asked whether synchrony across cholinergic interneurons varies as a function of movement state. At movement onsets, including both jerks and locomotion initiations, transients were highly synchronized (*Figure 4F,H,J*; *Figure 4—figure supplement 1*) and correlations across cell pairs were significantly larger than during rest (p<1e-90, Wilcoxon rank-sum test; *Figure 4H–I*; *Figure 4—figure supplement 1C*). Moreover, there was a significant negative correlation at onsets between cell-cell correlation and anatomical distance, not present during rest periods (p<1e-9, R = −0.26, *Figure 4H,J,L*; *Figure 4—figure supplement 1*). Importantly, correlations between cell pairs and number of activated neurons per transient were significantly lower during continuous running periods on average than at onsets or at rest (*Figure 4G–L*; *Figure 4—figure supplement 1C*), but like onsets, there was a strong negative relationship between cell pair correlation and distance (r = −0.42, p<1e-9; *Figure 4K*). Synchrony was lower during continuous locomotion despite larger overall mean DF/F relative to onsets or rest (*Figure 3F*). These results indicate that synchrony and functional spatial organization across ChIs is not fixed, but changes with an animal's movement state. The decrease in ChI population transient signaling measured with photometry during continuous running (*Figure 1D,E,G*) relative to locomotion onset is therefore not likely a result of decreased transient amplitudes in single ChIs (*Figure 3F*), but reflects decreased synchrony of transients (*Figure 4F–K*).

The data presented thus far provides evidence that sub-second timescale signals in striatum ChI networks vary with respect to changes in spontaneous movement states. Previous work has shown that SNc cell bodies, dopamine projection axons in dorsal striatum, and dopamine release rapidly fluctuate to spontaneous locomotor accelerations and decelerations and likely influence movement drive on sub-second timescales via rapid modulation of striatum output pathways (*Howe and Dombeck, 2016*; *da Silva et al., 2018*; *Patriarchi et al., 2018*; *Coddington and Dudman, 2018*; *Dodson et al., 2016*). Given the wealth of in-vitro and behavioral work that has pointed to direct and indirect interactions of the dopaminergic and cholinergic systems and their importance for proper regulation of movement (*Collins et al., 2016*; *Morris et al., 2004*; *Aosaki et al., 1994b*; *Di Chiara et al., 1994*; *Threlfell et al., 2012*; *Wang et al., 2006*; *Raz et al., 1996*; *Straub et al., 2014*; *Kosillo et al., 2016*), we aimed to determine, for the first time, how dorsal striatum dopamine and acetylcholine signaling co-vary with spontaneous changes in movement on sub-second timescales.

To measure simultaneous changes in ChI and dopamine axon signaling in the dorsal striatum of the same mouse we utilized a new bilateral fiber photometry recording strategy. We first established that signaling in dorsal striatum dopamine axons was highly synchronized across the two-hemispheres with zero phase lag (DAT-cre, n = 4; *Figure 5A,D,E*; mean phase lag DA v DA 0.6 ms, S.E.M 3.5 ms). We then determined that ChI population signals were similarly highly synchronized across the two-hemispheres of the dorsal striatum with zero phase lag (ChAT-cre, n = 6; *Figure 5B,F,G*; mean phase lag ChI vs ChI 0 ms, S.E.M 5.4 ms). These new findings indicate that DA and ChI population transients during locomotion likely reflect general changes in movement state, rather than the kinematics of specific limbs. Moreover, they provided the opportunity to simultaneously compare the relative timing and amplitude of the dorsal striatum DA signals in one hemisphere with the ChI

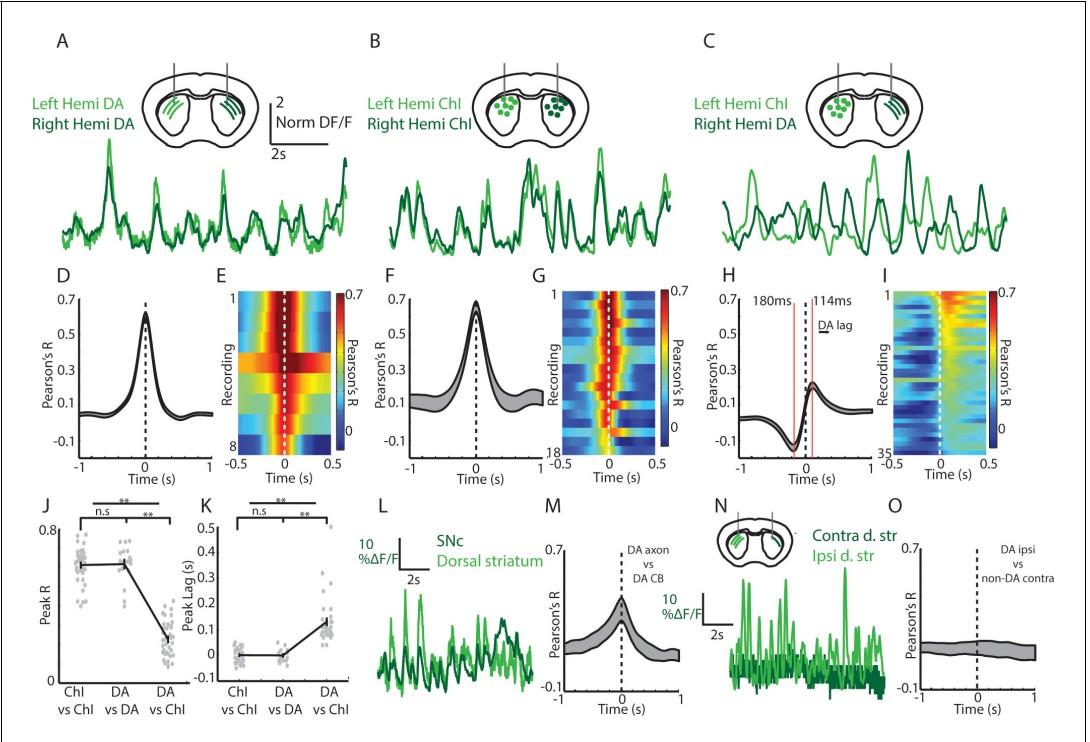

**Figure 5.** Cross-hemisphere synchrony permits simultaneous comparison of DA and ChI signals during spontaneous movement. (A) Representative z-score normalized population signals from GCaMP6f expressing DA axons recorded in the left and right hemispheres in the dorsal striatum of a DAT-cre mouse. (B) Representative bi-lateral population signals from dorsal striatum ChIs in a ChAT-cre mouse. (C) Representative population signals from GCaMP6f expressing ChIs recorded in the left striatum and DA axons in the right in a DAT-cre/ChAT-cre mouse. (D) Mean cross correlations between DA signals in the left and right dorsal striatum (n = 8 recording sites in 4 DAT-cre mice). (E) All cross-correlations between DA signals in left and right striatum. Each row corresponds to one recording. (F) Mean cross correlations between ChI signals in the left and right dorsal striatum (n = 18 recordings in 6 ChAT cre mice). (G) All cross-correlations between ChI signals in left and right dorsal striatum. (H) Mean cross correlations between dorsal striatum DA signals and ChAT signals in opposite hemispheres (n = 36 recordings in 8 DAT-cre/ChAT-cre mice). (I) All cross-correlations between ChI and DA signals in opposite hemispheres. (J) Mean peak correlation coefficients for cross-hemisphere population DA signals, ChI signals, and DA/ChI signals. Each dot represents a single recording. (K) Peak lag for cross-hemisphere population DA signals, ChI signals, and DA/ChI signals. Each dot represents a single recording. Positive lag for DA/ChI recordings indicates DA lag. (L) Representative normalized population signals from DA neurons recorded in the SNc and DA axon terminals in the ipsilateral striatum. (M) Mean cross-correlations between SNc and ipsilateral DA terminals in striatum (n = 16 recordings in three mice). (N) Representative population signals from DA axons in the striatum ipsilateral and contralateral to unilateral SNc DA neuron expression. (O) Mean cross-correlations between DA axons ipsilateral and contralateral to unilateral SNc DA neuron expression (n = 10 recordings in three mice). Shaded regions and error bars represent ±SEM. **p<1×10$^{-8}$; Wilcoxon Rank-Sum test.

DOI: https://doi.org/10.7554/eLife.44903.010

The following figure supplement is available for figure 5:

**Figure supplement 1.** Cross-hemisphere measures are consistent across expression configuration and are not a result of subcellular signaling differences.

DOI: https://doi.org/10.7554/eLife.44903.011

signals in the other by unilaterally expressing GCaMP6f in midbrain DA neurons and in ChIs in the opposite hemisphere (*Figure 5C,H,I*; DAT/ChAT cre mice, n = 8). Notably, peak cross-correlations between signals from dorsal striatum ChIs and contralateral DA axons was significantly lower than for DA/DA or ChI/ChI bilateral comparisons regardless of which hemispheres the ChI/DA axon signals were recorded (p<1e-10, Wilcoxon rank-sum test; *Figure 5J*, *Figure 5—figure supplement 1A*). In addition, the peak and trough in the DA/ChI cross correlations were significantly phase offset from 0 in all recordings (*Figure 5C,H,I,K*; mean peak lag +114 ms, mean trough lag −180 ms), regardless of hemisphere (*Figure 5—figure supplement 1B*). The peak in correlation at positive lag indicates an increase in dopamine lagging an increase in ChI signal. These timing offsets between ChIs and DA axon signals could not be attributed to different calcium dynamics between the subcellular compartments recorded (*Figure 5M*, *Figure 5—figure supplement 1C–D*). Simultaneous

recordings from dopaminergic SNc cell bodies and DA axons from the same hemisphere were correlated at near zero phase lag (*Figure 5L,M*) indicating that the DA/ChI timing differences are unlikely to be accounted for by axon transmission delays, which are estimated to be less ~15 ms for DA axons (*Pissadaki and Bolam, 2013*). Finally, very little signal and low correlations were detected from the striatum contralateral to the SNc injected hemisphere in DAT-cre mice injected unilaterally (*Figure 5N–O*; No ChI labeling), indicating little contribution from contralateral SNc projections or from other sources such as intrinsic fluorescence. These results establish a new approach for simultaneously monitoring DA and ChI signaling in striatum on sub-second timescales during behavior and indicate that DA axon terminals and ChIs exhibit different relative timing and amplitude variations relative to spontaneous movement.

We then asked how DA and ChI population signals co-vary depending on the animals' movement state in the absence of confounding changes in reward prediction or sensory stimuli. Simultaneous measurements of ChI and DA population signals were performed in mice (n = 8) that had never received rewards on the treadmill. Consistent with previous reports (*Howe and Dombeck, 2016*; *Patriarchi et al., 2018*), DA axon transient signaling, like ChIs, was stronger during movement periods relative to rest (p<1e-6 Wilcoxon rank-sum test, *Figure 6A,B*). Also, considered across all movement periods, both ChIs and DA axons were significantly correlated to acceleration and velocity but with different relative timing (*Figure 6C,D*). Movement onsets from rest (all jerks, initiations, and unclassified), on average, were associated with an increase in mean DA signaling relative to rest (*Figure 6E*; p<0.01 Wilcoxon rank-sum). This increase was smaller relative to that in ChIs (*Figure 6F*) because of both a higher DA signal at rest and a smaller average increase at onsets (*Figure 6E,F*). Unlike ChIs, however, DA axons exhibited elevated mean signaling during continuous locomotion relative to onsets (*Figure 6E*, p<0.01 Wilcoxon rank-sum). These data indicate that DA and ChIs both respond to spontaneous changes in movement, but their contribution to distinct movement states may not be equivalent.

We then examined whether ChI and DA axon signaling differed depending on the type of movement onset (to continuous locomotion or to a jerk). Both increased rapidly at the initiations of long locomotion bouts, with DA signaling slightly lagging ChIs (*Figure 6G,I*, *Figure 6—figure supplement 1C*). However, at the onsets of rapidly terminated jerks (initiation and termination within ~1 s), DA axon population signals rapidly decreased below rest baseline, while ChI populations increased to a level similar to locomotion bout initiations (*Figure 6G–J*, *Figure 6—figure supplement 1D*). Consistent with this finding, population ChI transients occurring at rest, while strongly predictive of subsequent transitions to movement (*Figure 6—figure supplement 1A*), were associated with significantly larger increases in velocity when they were accompanied by large DA axon signal increases (*Figure 6K,M*, *Figure 6—figure supplement 1E–F*). Interestingly, DA axon signaling at rest alone was only weakly predictive of transitions to movement (*Figure 6—figure supplement 1B*), and DA transients in the absence of accompanying ChI signal were not associated with significant increases in velocity above rest (*Figure 6L–M*, *Figure 6—figure supplement 1G–H*). These results indicate that DA and ChI signals can either converge or diverge with changes in movement from rest and may play complementary roles in controlling onsets. Synchronized ChI network transients signal a transition in the movement state (rest to move) while the subsequent DA signals indicate whether that movement is facilitated (DA increase, locomotion initiation) or rapidly terminated (DA decrease, jerk).

While DA and ChI signals correlated positively with each other at locomotion initiations and negatively at jerks (*Figure 7—figure supplement 1A*), continuous locomotion periods were associated with a strong negative correlation on average between DA axon and ChI population signals, indicating a ChI population decrease following a DA population increase (*Figure 7A–C*, *Figure 7—figure supplement 1A*). Both signals were correlated similarly to rapid changes in acceleration during locomotion (*Figure 7D*, indicating a slight lag between rapid acceleration and increases in ChI or DA signal). However, DA and ChI populations were correlated very differently with velocity: DA signaling showed a positive peak correlation with velocity at near zero phase lag, while ChI signals showed a negative trough in the correlation at positive lags (*Figure 7E*). These findings indicate that velocity increases during continuous locomotion were associated with a coordinated increase in DA signal followed by a decrease in ChI population synchrony (*Figures 4* and *7F–G*). Conversely, decreases in velocity were associated with decreases in DA signal, followed by increases in ChI synchrony (*Figure 7F,H*). When synchronous ChI population transients occurred during continuous locomotion,

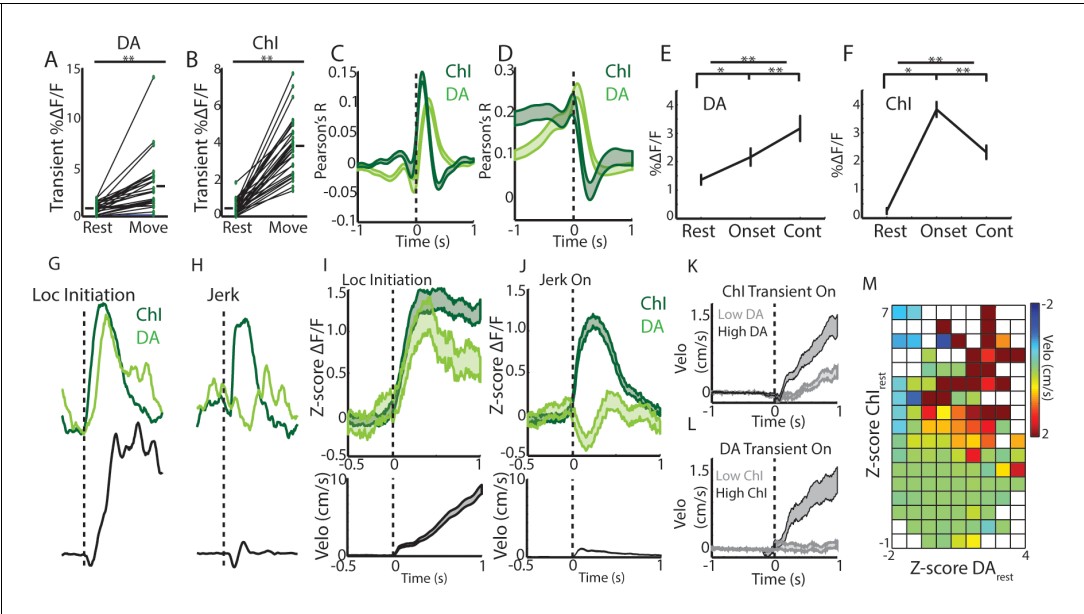

**Figure 6.** Rapid, simultaneous DA terminal and ChI signals at rest exhibit both parallel and divergent dynamics which co-vary with movement onset. (**A**) Mean transient DF/F for all dorsal striatum DA axon terminal recordings (n = 25 recordings, eight mice) recorded simultaneously with ChIs during movement and rest periods. (**B**) Mean transient DF/F for ChI populations recorded simultaneously with DA terminals. (**C**) Mean cross-correlations between DF/F and acceleration for all simultaneously recorded DA terminals and ChIs. (**D**) Mean cross-correlations between DF/F and velocity for all simultaneously recorded DA terminals and ChIs. (**E**) Mean DA terminal DF/F during all rest, onset, and continuous locomotion periods. (**F**) Mean ChI population DF/F during all rest, onset, and continuous locomotion periods. (**G**) DF/F from simultaneously recorded ChIs and DA terminals aligned on a representative locomotion initiation. (**H**) DF/F from simultaneously recorded ChIs and DA terminals aligned on a representative jerk onset. (**I**) Mean z-score normalized DF/F for all simultaneous DA terminal and ChI recordings (top) aligned on locomotion initiations (velo, bottom; n = 94 initiations, eight mice). (**J**) Mean z-score normalized DF/F for all simultaneous DA terminal and ChI recordings (top) aligned on jerk onsets (velo, bottom; n = 292 jerks, eight mice). (**K**) Velocity triggered on the onsets of ChI population transients occurring at rest which were associated with high (n = 525 transients, top quartile) and low (bottom quartile) DA terminal signals (see *Figure 6—figure supplement 1* for associated DF/F traces). (**L**) Same as K, for onsets of DA terminal transients (n = 319 for top and bottom quartile of associated ChI DF/F). (**M**) Z-score normalized ChI and DA terminal population DF/F for each significant transient event occurring at rest and the mean change in velocity in a 1 s period post transient onset for each combination of DA and ChI values. Shaded regions and error bars represent ±SEM. **p<1×10$^{-8}$; Wilcoxon Rank-Sum test.

DOI: https://doi.org/10.7554/eLife.44903.012

The following figure supplement is available for figure 6:

**Figure supplement 1.** DA terminal and ChI signals at rest are associated with distinct aspects of transitions to movement.
DOI: https://doi.org/10.7554/eLife.44903.013

they were associated with rapid decreases in velocity when not accompanied by strong DA signals, but were associated with increases in velocity or 'resets' in locomotion when DA and ChI population signals occurred in conjunction (*Figure 7A,I–J, Figure 7—figure supplement 1B–E*). At locomotion terminations to rest, relative ChI population signaling increased slightly prior to movement cessation, while DA signals decreased with the drop in velocity (*Figure 7K*). These data indicate that fluctuations in running speed during continuous locomotion are associated with divergent patterns of dorsal striatum ChI and DA network activity. Increases in the speed or vigor of ongoing locomotion were associated with increases in DA population signals followed by drops in ChI population synchrony associated with movement state transitions. Velocity decreases, in contrast, were marked by coincident decreases in DA population signal, followed by increases in ChI synchrony as movement state transitions became more likely (locomotion resets and terminations). Thus, ChI population transients may mark movement state transitions, which are either enacted and invigorated (DA increase) or terminated (DA decrease).

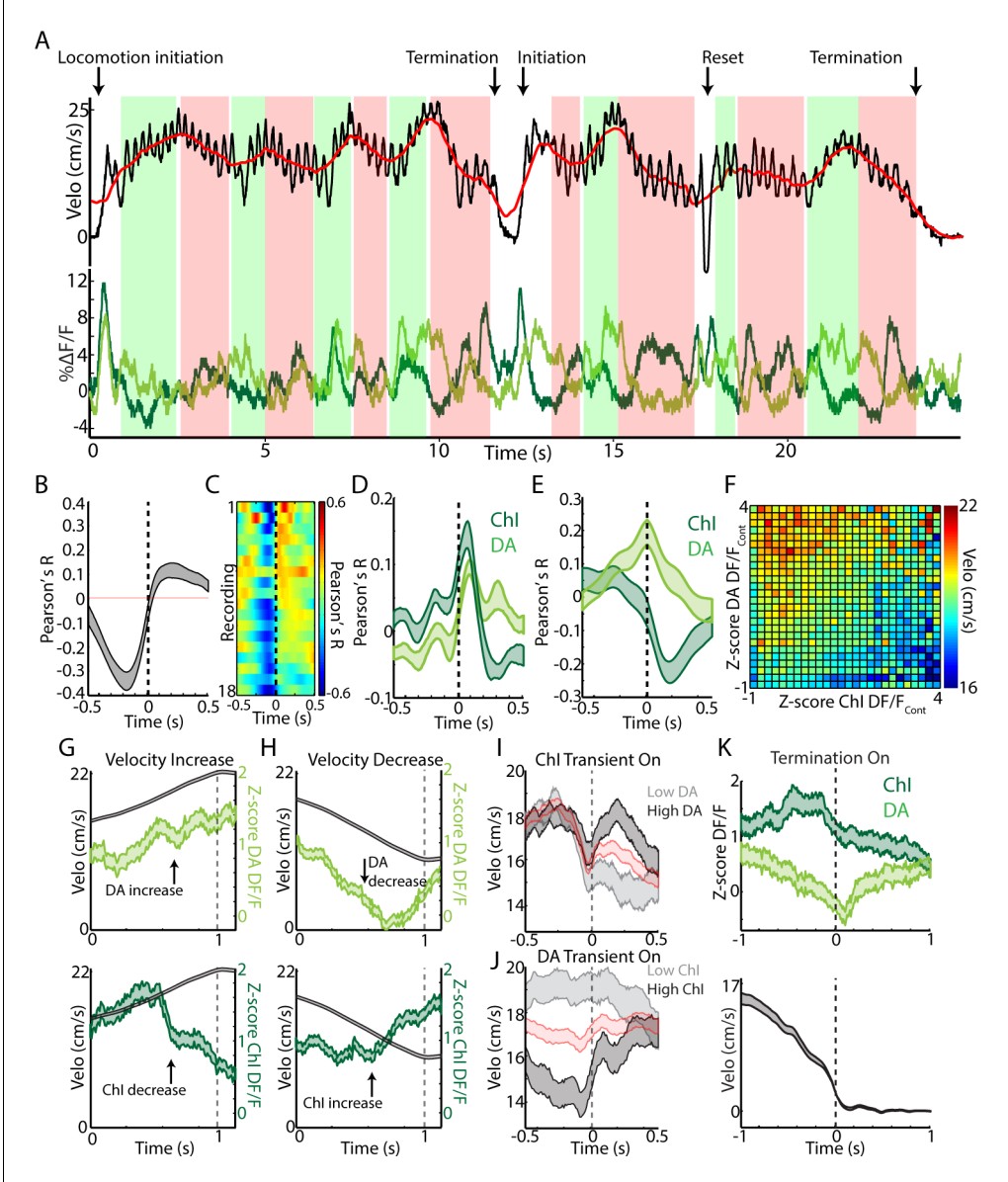

**Figure 7.** Changes in velocity during continuous locomotion are associated with divergent DA and ChI population signals. (**A**) Velocity (top) and simultaneously recorded ChI and DA terminal population signals (bottom) during a representative period of continuous locomotion. Red line indicates smoothed velocity showing fluctuations in running velocity superimposed on rapid cyclic accelerations. Green shaded regions indicate velocity increases, red decreases. (**B**) Mean cross correlation between DA terminals and ChIs during continuous locomotion (n = 18 recordings, eight mice). (**C**) Cross correlation between DA and ChIs for each recording, sorted by peak correlation magnitude. (**D**) Mean cross correlation between ChI and DA populations and acceleration during continuous locomotion. (**E**) Mean cross correlation between ChI and DA populations and velocity during continuous locomotion. (**F**) Z-score normalized ChI and DA terminal DF/F and the associated mean velocity in the same time bins for each combination of DA and ChI values during continuous locomotion. (**G**) Mean velocity (black) and DA (top) and ChI (bottom) DF/F aligned on the peak of velocity increases during continuous locomotion (n = 451). (**H**) Same as G aligned on the trough of velocity decreases during continuous locomotion (n = 724). (**I**) Mean velocity aligned on the onsets of significant population ChI transients during continuous locomotion associated with high (top quartile, n = 155) or low (bottom quartile, n = 155) mean DA signal. Red trace is mean across all (n = 610 transients). See *Figure 7—figure supplement 1* for DF/F traces. (**J**) Same as I for DA population transients (n = 426) associated with high (n = 106 transients) and low (n = 106) ChI signal. (**K**) Mean ChI and DA DF/F (top) and velocity (bottom) aligned on terminations of continuous locomotion to rest (n = 83, eight mice). Shaded regions and error bars represent ±SEM.

*Figure 7 continued on next page*

*Figure 7 continued*

DOI: https://doi.org/10.7554/eLife.44903.014

The following figure supplement is available for figure 7:

**Figure supplement 1.** Summary of co-varying DA and ChI signals during spontaneous movement.

DOI: https://doi.org/10.7554/eLife.44903.015

## Discussion

Clinical and behavioral studies have implicated striatal cholinergic interneurons as powerful modulators of movement control. This regulation is believed to occur through the complex action of acetylcholine on striatal projection neurons and via direct and indirect interactions with the nigrostriatal dopamine system. Despite the clear importance of striatal cholinergic and dopaminergic signaling for normal movement, the naturally occurring signaling patterns of cholinergic interneurons and their relationship to dopaminergic signals during ongoing behavior have remained unclear. Using a combination of cellular resolution imaging and population recording approaches with genetically encoded calcium indicators in behaving mice, we have shown that dorsal, but not ventral, striatum cholinergic interneuron populations exhibit rapid, transient signaling locked to transitions in spontaneous movement. These population signals were the result of widespread transient synchronization of ChIs across large regions of the dorsal striatum, which dropped off independently of single cell activity after successful transitions to continuous locomotion. By measuring cholinergic and dopaminergic signals simultaneously, we established that these neuromodulator systems, while both related at a gross level to rapid movement changes, displayed both coordinated and divergent signals depending on the nature of the movement. This data supports a model in which dopamine and acetylcholine are not purely competitive, but can participate both independently and cooperatively in ongoing movement control, with cholinergic population synchrony signaling rapid changes in movement states and dopamine signaling providing rapid drive to enact and sustain new movement states (*Figure 7—figure supplement 1F*).

### Transient cholinergic interneuron population synchrony marks movement state transitions

Transient increases in ChI population synchrony were associated with transitions from rest to movement and 'resets' in ongoing locomotion (*Figure 1*). It is likely given the wide variation of the treadmill movement profiles (jerks, locomotion initiations, resets; *Figure 1*) and the strong population synchrony across hemispheres (*Figure 5B*) that ChI population transients in this study were related to non-specific changes in movement state involving the general activation of multiple limbs and muscle groups. In further support of this idea is the observation that transient synchrony dropped off dramatically as mice ran at higher velocities, despite similar limb recruitment (*Figures 1*, *4* and *7*). Movements requiring lateralized body part activation (such as turns) may recruit asymmetric patterns of cross hemisphere activation, as for striatal principle neurons (*Cui et al., 2013*), but the movements tested here likely involved similar bi-lateral limb recruitment. ChI responses to movements were expressed in the absence of experimentally generated salient or reward predictive cues, rewards, or task contingencies, establishing them as uniquely related to changes in internally triggered spontaneous movement. Prior studies in primates have described very little response in TANs to movements per se (*Apicella et al., 1996*; *Apicella et al., 1997*; *Apicella et al., 1991*; *Kimura et al., 1984*; *Ravel et al., 1999*; *Ravel et al., 2003*; *Morris et al., 2004*; *Aosaki et al., 1994b*; *Nougaret and Ravel, 2015*; *Matsumoto et al., 2001*; *Apicella, 2002*), while studies in rodents described some TAN responses to task-related movements (*Yarom and Cohen, 2011*; *Benhamou et al., 2014*). One possibility for this species discrepancy is the nature of the movement types being performed (cue triggered single arm or eye movements vs internally triggered whole body locomotor movements). Also, movement responses may be less apparent in firing rate changes of single cells relative to across large synchronous populations. We hypothesize that synchronous ChI network activations support rapid changes or reconfigurations in the striatum network that can facilitate general, internally triggered changes in movement state (*Figure 7—figure supplement 1F*). This idea is consistent with the proposed role for ChI pauses (and rebound bursts) in facilitating

stopping or attentional orienting to salient external cues (*Apicella et al., 1996*; *Kondabolu et al., 2016*; *Schulz et al., 2011*; *Thorn and Graybiel, 2010*).

## Relationship of dorsal ChI signals to stimuli and rewards

Classic in vivo electrophysiology studies in primates have identified responses in putative striatum ChIs (TANs) to salient, aversive, and reward predictive stimuli and to primary rewards and punishments (*Apicella et al., 1996*; *Apicella et al., 1997*; *Apicella et al., 1991*; *Kimura et al., 1984*; *Ravel et al., 1999*; *Ravel et al., 2003*; *Morris et al., 2004*; *Aosaki et al., 1994b*; *Nougaret and Ravel, 2015*; *Matsumoto et al., 2001*; *Apicella, 2002*). These are often composed of a short latency burst followed by a pause, then a longer latency 'rebound' excitation. Because of the difficulty in accurately distinguishing ChIs from other striatal cell types using in vivo recording, responses to stimuli and rewards have yet to be confirmed in genetically identified ChIs of behaving rodents or primates. In dorsal striatum ChI populations, we report the presence of a short latency increase, followed by a decrease to unpredicted water rewards (*Figure 2D*), which was even more pronounced to reward predictive stimuli (*Figure 2—figure supplement 1C,D,G*). We found that the relative magnitudes of the bursts and pauses were strongly related from trial to trial to changes in movement of the mouse at the time of unpredicted reward delivery (*Figure 2F,G,H*) but not at the time of the predictive conditioned stimulus (*Figure 2—figure supplement 1C,D,G*). Thus, signals in dorsal striatum ChIs likely reflect a combination of sensory stimuli and movements which may be integrated to either rapidly influence movement output or to drive longer timescale plasticity.

## Differences in ChI signaling across the dorsal-ventral axis of the striatum

We measured marked differences in population ChI responses across the dorsal-ventral axis. While dorsal striatum ChIs increased signaling rapidly at movement initiations and locomotion resets, ChI populations in ventral striatum (NAc core), did not exhibit rapid changes at spontaneous movement initiations prior to introduction of rewards, but developed weak responses to movement initiations after rewards were introduced, perhaps reflecting changes in reward prediction (*Figure 2B–C*, *Figure 2—figure supplement 1A*). Large transients were present in ventral ChIs to later introduced unpredicted water rewards and reward-predictive cues independently of movement, consistent with some prior ventral striatum TAN recordings in rodents (*Atallah et al., 2014*; *Benhamou et al., 2014*) (*Figure 2—figure supplement 1E–G*). In primates, TAN cue and reward responses are more pronounced in ventral striatum relative to dorsal, but unlike the monophasic increases reported here in ventral ChIs, these were primarily pauses (*Marche et al., 2017*). It is possible that rapid pauses in ongoing firing may be more difficult to detect with calcium indicators. However, we were able to detect rapid decreases below baseline in dorsal ChI populations to conditioned cues (*Figure 2—figure supplement 1C,D,G*). The dorsal-ventral gradient in unpredicted reward and spontaneous movement signaling in ChIs resembled that previously reported for dopamine axons in the striatum (*Howe and Dombeck, 2016*; *Parker et al., 2016*). Thus, neuromodulator systems exhibit parallel functional differences across the dorsal ventral striatum axis, in line with proposed functional roles of the dorsal and ventral striatum in movement control and reward-learning respectively.

## Synchrony across ChIs determines the population signal and varies with behavior

Two-photon imaging of populations of dorsal striatum ChIs was performed to determine how movement related signals were expressed across networks of single neurons. ChIs exhibited widespread, highly synchronous transients at movement onsets involving neurons separated by 100 s of microns (*Figure 4*). This high degree of network synchrony dropped off with distance between cells and is consistent with recordings from pairs of TANs showing synchronous pauses to stimuli (*Morris et al., 2004*). Interestingly, synchrony across ChIs decreased significantly below baseline resting levels following transitions to ongoing locomotion (*Figure 4H,I*), while single cell transient amplitude and frequency did not significantly decrease (*Figure 3F*). Thus, synchrony is not a fixed property of cholinergic networks but varies dynamically with respect to ongoing behavior. The degree of synchrony, but not the magnitude of signals from individual cells, corresponds with the prominent transients measured by photometry, an observation that may apply generally to bulk photometry

recordings (*Figure 1D*). In this case, summation of asynchronous transients from single ChIs during continuous locomotion would not result in detectable transient changes in population photometry measures. Widespread synchrony in ChI networks is an important determinant of nicotinic receptor mediated pre-synaptic dopamine release from axon terminals and could impact oscillatory network dynamics and enhance the strength of post-synaptic cholinergic modulation (*Kondabolu et al., 2016*; *Lim et al., 2014*; *Threlfell et al., 2012*).

## Coordinated and divergent cholinergic and dopaminergic movement signaling in dorsal striatum

Interplay between dopaminergic input from SNc and local cholinergic modulation in striatum has long been hypothesized to play a role in the proper control of movement drive. A leading model is that dopamine and acetylcholine compete for ongoing movement control, where dopamine is pro-kinetic and acetylcholine is anti-kinetic. Disruption of this balance in favor of acetylcholine is believed to contribute to akinesias observed in Parkinson's Disease, a hypothesis consistent with clinical and behavioral studies. Mechanistic studies have provided a more complex, often contradictory, view of cholinergic circuit effects on multiple time-courses and spatial scales (*Lim et al., 2014*; *Oldenburg and Ding, 2011*; *Threlfell et al., 2012*; *Thorn and Graybiel, 2010*) which does not strictly support a purely anti-kinetic, dopamine competition model. To clarify how dynamic DA and Ach interactions contribute to movement drive, we measured simultaneous calcium transients from ChI populations and dopaminergic axons in the striatum during spontaneous, unrewarded locomotor movements (*Figures 5–7*). We found that, while both displayed rapid signaling during changes in movement, DA and ChI population signaling deviated significantly during certain movement phases, but were coordinated during others. These results suggest a new model in which DA and ChI signals are associated with different aspects of movement and may act both cooperatively and competitively depending on when signals are expressed (*Figure 7F*).

Recent studies on the activation of DA neurons and terminals, have provided evidence for movement related activation and inhibition under different task conditions (*Howe and Dombeck, 2016*; *da Silva et al., 2018*; *Coddington and Dudman, 2018*; *Dodson et al., 2016*). Our results suggest that these seemingly conflicting findings may be partly explained by differences in the types of movements examined. Here we show that dorsal striatum DA terminal signals, in the absence of learned reward contingencies, increased at transitions to locomotion but decreased below rest levels at the onset of short, rapidly terminated movements (jerks; similar to many of the short movements examined recently (*Coddington and Dudman, 2018*; *Dodson et al., 2016*); *Figure 6*). This pattern contrasted with ChI populations, which increased synchronous signaling just prior to any transition to whole-body treadmill movement, regardless of whether locomotion was initiated (*Figure 6*). Because of this bi-directional DA activation and the presence of spontaneous DA transients during rest, DA signals were less predictive (though still predictive) of general movement onsets than the ChI population synchrony transients (*Figure 6E–F,I–M*, *Figure 6—figure supplement 1A–B*). Thus, although optogenetic stimulation of DA projections to dorsal striatum can promote movement initiations (*Howe and Dombeck, 2016*; *da Silva et al., 2018*; *Barter et al., 2015*), DA transients *alone* are likely not sufficient to trigger movement initiations (*Howe and Dombeck, 2016*; *Coddington and Dudman, 2018*; *Saunders et al., 2018*). DA driven invigoration of locomotion likely occurs when the DA stimulation is properly timed with other striatal signals, such as synchronous ChI transients or activation of projection neurons, which directly promote changes in movement state. Such an interpretation is consistent with the on-average increase in rapid locomotion initiation probability we observed previously in response to DA stimulation, and also consistent with the trial-by-trial and session-by-session variability in the effectiveness of the stimulation to promote locomotion observed in the same study (*Howe and Dombeck, 2016*).

We propose that rapid DA signals at movement onsets signal the drive to enact or invigorate a triggered movement (movement initiation → locomotion) or to increase or decrease the vigor of that movement once initiated (*Figure 7—figure supplement 1F*). Synchronous ChI network transients at rest, on the other hand, signal rapid changes in the current movement state (rest → movement initiation) and transients after initiation signal resets in ongoing locomotion or termination (*Figure 7—figure supplement 1F*). Similar to dopamine, the causal impact of synchronous ChI signals on behavior is likely not purely deterministic in isolation, but could participate in conjunction with coincident dopaminergic, glutamatergic, and local GABAergic inputs to modulate the final

output from SPN populations to facilitate movement transitions. While these DA terminal and ChI population signals are clearly subject to independent modulation in part, perhaps through different sets of inputs, the close correspondence of ChI and DA terminal increases at locomotion initiations suggest the possibility for direct feedforward amplification of DA terminal signals by synchronous ChI networks, as described in vitro (*Threlfell et al., 2012*). An alternative hypothesis is that transient ChI signals are anti-kinetic and compete with pro-kinetic DA signals for movement control. However, this is not consistent with our observation of large ChI signals associated with locomotion onsets (*Figure 6*). Future targeted manipulation studies and simultaneous recordings will be required to elucidate the causal mechanisms responsible for the observed dynamics and their impact on striatal circuitry and behavioral output.

This framework of DA and ChI signaling during movement provides a dynamic foundation for interpreting studies implicating dorsal striatum acetylcholine in set switching and orienting to cues (*Apicella et al., 1996*; *Kimura et al., 1984*; *Ding et al., 2010*; *Aoki et al., 2015*) and with dopamine in regulating movement vigor (*da Silva et al., 2018*; *Panigrahi et al., 2015*; *Niv et al., 2007*). However, it is seemingly at odds with studies showing pro-kinetic effects of cholinergic block on Parkinson's symptoms and the anti-kinetic effects of cholinergic neuron stimulation on gross locomotion (*Kondabolu et al., 2016*; *Barbeau, 1962*; *Bordia et al., 2016b*; *Maurice et al., 2015*; *Ztaou et al., 2016*). One possibility is that synchronous ChI signaling in the absence of DA signaling becomes anti-kinetic because it promotes movement state changes without the accompanying movement drive. DA signaling following movement initiations may facilitate smooth transitions to locomotion, in part, by desynchronizing the ChI networks (*Raz et al., 1996*), reducing the probability of changing or terminating movement after initiation. Without ChI desynchronization and the drive to invigorate the new movement state provided by DA, artificial stimulation of ChI cells (with optogenetics) may induce continuous state changes such as slow, jerky movements if the stimulation is delivered at rest or stopping if delivered during locomotion. Similarly, Parkinson's patients may exhibit jerky, bradykinetic locomotor patterns in part because synchronized ChI signaling is not accompanied by DA signals which desynchronize (change promoting) ChI networks and enable transitions to stable continuous movement.

## Materials and methods

### Stereotaxic virus injections

All experiments were approved by the Northwestern University Animal Care and Use Committee. Animal numbers used for each experiment and analysis are indicated in figure legends and main text. Heterozygous adult male mice (postnatal 3–6 months, n = 20) with Cre expression in ChAT containing cholinergic neurons (B6;129S6-*Chat*<sup>tm2(cre)Lowl</sup>/J; Jackson Labs) were anesthetized with isofluorane (1–2%) and a 0.5–1 mm diameter craniotomy was made over the right and/or left striatum (+0.5 mm rostral,±1.8 lateral from bregma). Flexed GCaMP6f virus (AAV1–Syn–flex–GCaMP6f (*Chen et al., 2013*), $1.4 \times 10^{13}$ GC ml$^{-1}$ diluted 1:1 in PBS; University of Pennsylvania vector core) was pressure injected through a pulled glass micro-pipette at four depths in the striatum (−1.6,–1.9, −2.2,–2.5). 150 nl was injected at each depth for a total of 0.6 µL per hemisphere. A subset of mice (n = 8, *Figure 2*) also received injections into the ventral striatum at four depths (−3.6,–3.9, −4.2,–4.5 mm). In some 2-photon imaged mice, virus was injected at two rostral/caudal locations (+0.6, +0.3 anterior) and two depths (−1.6 mm and −1.9 mm). For bilateral DA terminal and cholinergic population recordings (*Figures 5–7*, n = 8), ChAT-cre mice were crossed with DAT-cre mice (B6.SJL-*Slc6a3*<sup>tm1.1(cre)Bkmn</sup>/J; Jackson Labs) to generate a hybrid DAT-cre/ChAT-cre strain. One hemisphere was injected in the striatum as above to drive expression in cholinergic interneurons, and the other was injected at two locations and three depths in the midbrain SNc (−3.1 and −3.4 caudal,+1.4 mm lateral, −3.8,–4.1,–4.4mm; 100 nL/depth) to drive expression in dopaminergic cells. Following the injections, the skull and craniotomy were sealed with Metabond (Parkell) and a custom metal headplate. Two-photon and fiber photometry recordings were made 3–5 weeks post-injection.

### Behavior

Mice were head-fixed with their limbs resting on a 1D cylindrical styrofoam treadmill ~8in in diameter by 5in wide (treadmill described previously (*Howe and Dombeck, 2016*)) in the dark, which

allowed them to run freely forwards and backwards. Rotational velocity of the treadmill was sampled at 1000 Hz by a rotary encoder (E2-5000 US Digital) attached to the axel. All mice were acclimated to the treadmill for 2–4 days until they exhibited consistent, spontaneous transitions between movement (jerks or continuous locomotion) and rest. A subset of mice (*Figure 2*) were placed on water scheduling (1 mL daily) and received randomly delivered water rewards (8–10 uL) through a spout placed near their mouths that were non-contingent upon stimuli or movements. Another subset (*Figure 2—figure supplement 1C–G*, n = 4) was trained on a delay classical conditioning task where a 3 s blue light stimulus (delivered at random intervals, 10 s-30s uniform distribution) was paired with water reward with 100% probability. Recordings were made after mice exhibited consistent pre-reward spout licking (measured with a contact monitoring circuit) in response to the stimulus (*Figure 2—figure supplement 1G*).

## Fiber photometry

The day prior to acute fiber photometry recordings, mice were anesthetized under isofluorane (1–1.5%) and a hole was drilled in the Metabond covering the injected striatum. Bone was removed and the exposed brain surface was covered in Kwik-Sil (WPI). On the day of recording, the Kwik-Sil was removed and 200 µm diameter optical fibers (two fibers for bilateral recordings, Doric) were slowly lowered into the dorsal striatum (+0.5 mm rostral,±1.8 mm lateral from bregma) using a table mounted micro-manipulator (Sutter) while mice were head-fixed on the treadmill. The custom designed fiber photometry system was previously described (*Howe and Dombeck, 2016*). Briefly, excitation light (488 nm diode laser, 0.6–0.8 mW power at the fiber tip) was delivered continuously and emitted population GCaMP6f fluorescence was filtered and measured with a GaAsP PMT (H10770PA-40, Hamamatsu). The PMT output was sampled at 1 kHz and synchronized with the velocity output from the rotary encoder using an oscilloscope and associated software (Picoscope 4824). Continuous recordings in the dorsal (1.7–2.1 mm depth from surface) or ventral striatum (3.8–4.4 mm depth) were carried out over multiple depths (1-3) in a given day (1–3 days/mouse) and each comprised a 3–6 min behavior period. Not all recordings met behavior criteria to be included in all analyses (see Data Analysis). After each striatum recording session, the fibers were slowly raised into the overlying cortex (0.5 mm from surface) where a final recording was made to determine the baseline tissue and optical fiber autofluorescence (see Data Analysis).

## Two-photon imaging

Mice for two-photon imaging of cholinergic interneurons (*Figures 3–4*) were implanted with a window cannula above the dorsal striatum (+0.5 mm rostral,+1.8–2.1 mm lateral from bregma) as previously described (*Howe and Dombeck, 2016*). After recovery from surgery, mice were water scheduled and acclimated to the treadmill. Imaging began after the window surface had cleared of any blood and cellular debris (typically 5–7 days post-surgery). Imaging was performed by a custom, table-mounted two photon microscope (*Heys et al., 2014*). Laser power after the objective (Olympus LUMPlanFL N, 40X, 0.8 NA and UMPLFLN, 20X, 0.5 NA) ranged between 50 and 150 mW (920 nm or 800 nm for control *Figure 3—figure supplement 1G*), but this was likely significantly lower at the sample because of clipping of the excitation light by the cannula. A Digidata1440A (Molecular Devices) data acquisition system was used to record (Clampex 10.3) and synchronize reward timing, licking, wheel velocity, and two-photon image frame timing. Time series movies (5,000–15,000 frames) were acquired at 30 Hz (1024 or 512 × 512 pixels) and field sizes ranged from 100 to 700 µm in diameter. Fields ranged in depth from 20 to 100 µm below the striatum surface. ROIs corresponding to somata and dendrites of cholinergic interneurons were hand-selected if they could be clearly discriminated from surrounding neuropil (*Figure 3A*). Images were obtained from an approximately 1.4 mm diameter region around the center of the cannula. Each mouse was imaged for 2–5 days with data from 2 to 4 fields acquired each day.

## Data analysis

*Baseline correction and pre-processing.* All data were analyzed using ImageJ (1.46) and custom functions written in MATLAB (Version 2012b). Fluorescence time series from fiber photometry were binned into 10 ms bins (100 Hz sampling) and raw fluorescence was converted to DF/Baseline F. A baseline for each bin was initially calculated as the 8th percentile fluorescence over a 16 s sliding

window around the bin, excluding fluorescence values above a threshold visually determined for each session to include significant positive-going transients. Each baseline value was then corrected to account for the fiber and tissue autofluorescence measured from the cortex in that same session (BaselineF - F from autofluorescence). Significant positive-going transients were identified by calculating the minimum duration for positive events of varying standard deviation magnitudes (1–6 + SD above mean) at which the fraction of positive to negative events exceeded 0.99 (p<0.01 probability of event resulting from random noise fluctuations or movement artifact induced false transient). These duration thresholds were calculated across all photometry recordings, then applied individually to each session. The same procedure was used to identify significant positive-going transients for each single-cell ROI in our 2-photon datasets. Z-score normalization (*Figures 6I–J* and *7G–H, K*) was calculated relative to the mean and standard deviation across all rest bins (see below) for each session.

## Classification of spontaneous treadmill behavior

All sessions and mice were included in behavioral analyses unless they did not contain treadmill movements meeting the criteria described here. The number of mice and sessions is indicated in the main text and Figure legends for each analysis. Acceleration was calculated as the difference in consecutive velocity measures, where the velocity was first smoothed with a 50 ms sliding window. For general quantification of fluorescence changes during rest and movement periods (*Figures 1C*, *3E* and *6A–B*), each bin was classified as occurring during a movement period associated with significantly elevated velocity and acceleration (velocity exceeding 2.2 cm/s and acceleration exceeding 40 cm/s$^2$ in a ± 1 s window), a rest period (no velocity exceeding 0.2 cm/s in a ± 1 s window), or neither/unclassified. These thresholds were chosen from visual inspection of the mouse velocity traces and were conservative estimates. General movement onsets from rest (*Figures 1E*, *2B* and *3G*; *Figure 1—figure supplement 1G*; *Figure 2—figure supplement 1A*, *Figure 3—figure supplement 1C*) were classified as positive-going threshold crossings of velocity at 0.2 cm/s where the peak velocity reached at least 1 cm/s within one second post-onset. To reduce the possibility of small movements preceding the onset, no absolute velocities exceeding 0.2 cm/s could be present within a 0.5 s period prior to the movement onset. Micro-movements falling below this threshold did not elicit detectable changes in the ChI signal and likely fell in the noise range of our rotary encoder (*Figure 1—figure supplement 1E*). Jerks (or rapidly terminated movement onsets; *Figures 1H,L* and *6J*; *Figure 4—figure supplement 1B*, *Figure 6—figure supplement 1D*) were classified as movement onsets from rest in which the maximum velocity in a window 1–2 s after the velocity threshold crossing was less than 2.2 cm/s (signaling a rapid return to rest). Locomotion initiations (*Figures 1G–K* and *6I*) were classified as movement onsets in which the mean velocity in a window 0.5 s-2s post onset was greater than 4.5 cm/s. Other unclassified onsets that did not meet these criteria were present, but these classifications reliably captured two common and distinct movement types observed by visual inspection (*Figure 1B*). Continuous locomotion periods (*Figures 1D*, *4G, K* and *7*; *Figure 7—figure supplement 1*) were classified as bouts lasting >3 s in which the mouse velocity was sustained above 4.5 cm/s. Locomotion terminations were classified as periods were mice crossed from a period of continuous locomotion below 2.2 cm/s then remained below 2.2 cm/s for the subsequent 1 s. For behavior specific correlations (*Figures 4* and *7B*; *Figure 4—figure supplement 1*, *Figure 7—figure supplement 1A*), data in a window from 0.1 to 2 s around each onset or during each continuous locomotion bout was concatenated and the cross correlation computed for each recording. Recording sessions with less than 5 s of total time for each analyzed behavioral epoch were omitted. Mean DF/F for specific behavioral periods was calculated using the same windows.

## Histology

Mice were perfused trans-cardially with 15 mL of PBS (Fischer) and 15 mL of 4% paraformaldehyde (EMS). Brains were stored in PBS at 4°C then transferred to 40% sucrose (Fischer) overnight before sectioning. Coronal slices (40–50 μm) were cut on a freezing microtome and stored at 4°C in PBS. For immunostaining of cholinergic interneurons, sections from a subset of mice were blocked in 5% serum, incubated overnight at 4°C with antibodies for GFP (Abcam, 1:1000 dilution) and choline acetyltransferase (Millipore, 1:500), then incubated 2 hr at room temperature with secondary antibodies

tagged with Alexa 488 (JacksonLabs, 1:250) and Alexa 647 (Invitrogen, 1:250) respectively. Imaging of ChAT and GCaMP6f expression was performed on an Olympus Slide Scanner (VS120) microscope (*Figure 1—figure supplement 1A*).

## Acknowledgements

We thank Drs. Ann Graybiel, Jill Crittenden and Jim Heys for comments on this manuscript. We thank the Kozorovitskiy Lab for use of their slide scanner.

## Additional information

### Funding

| Funder | Grant reference number | Author |
| --- | --- | --- |
| National Institutes of Health | R01MH110556 | Daniel A Dombeck |
| McKnight Foundation | | Daniel A Dombeck |
| National Institutes of Health | T32 AG20506 | Mark Howe |

The funders had no role in study design, data collection and interpretation, or the decision to submit the work for publication.

### Author contributions

Mark Howe, Conceptualization, Data curation, Formal analysis, Investigation, Methodology; Imane Ridouh, Anna Letizia Allegra Mascaro, Alyssa Larios, Data curation, Methodology; Maite Azcorra, Data curation, Formal analysis, Writing—review and editing; Daniel A Dombeck, Conceptualization, Supervision, Funding acquisition, Methodology, Writing—review and editing

### Author ORCIDs

Mark Howe https://orcid.org/0000-0002-6865-5084
Anna Letizia Allegra Mascaro https://orcid.org/0000-0002-8489-0076
Daniel A Dombeck https://orcid.org/0000-0003-2576-5918

### Ethics

Animal experimentation: All experiments were approved by the Northwestern University Animal Care and Use Committee (Protocol #IS00005043 and IS00003736).

### Decision letter and Author response

Decision letter https://doi.org/10.7554/eLife.44903.020
Author response https://doi.org/10.7554/eLife.44903.021

## Additional files

### Supplementary files

• Transparent reporting form
DOI: https://doi.org/10.7554/eLife.44903.016

### Data availability

Data generated or analysed during this study are included in the manuscript and supporting files. Processed data from large data files (time-series movies) that support the findings of this study are available at Dryad Digital Repository (doi:10.5061/dryad.244nt37).

The following dataset was generated:

| Author(s) | Year | Dataset title | Dataset URL | Database and Identifier |
|---|---|---|---|---|
| Ridouh I, Howe MW | 2019 | Data from: Coordination of rapid cholinergic and dopaminergic signaling in striatum during spontaneous movement | https://dx.doi.org/10.5061/dryad.244nt37 | Dryad Digital Repository, 10.5061/dryad.j1fd7 |

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
