## [Decision Letter]

Thank you for submitting your article "Coordination of rapid cholinergic and dopaminergic signaling in striatum during spontaneous movement" for consideration by *eLife*. Your article has been reviewed by three peer reviewers, and the evaluation has been overseen by a Reviewing Editor and Laura Colgin as the Senior Editor. The reviewers have opted to remain anonymous.

The reviewers have discussed the reviews with one another and the Reviewing Editor has drafted this decision to help you prepare a revised submission.

Summary:

Previous studies have indicated critical roles of cholinergic interneurons (ChIs) in the striatum in movement and learning. However, how ChIs regulate movement remain elusive. To address this issue, Howe and colleagues examined the activity of ChIs in the striatum during spontaneous movements. The authors used fiber photometry and single cell 2-photon calcium imaging to monitor the activity of ChIs. The authors show that ChIs exhibit synchronous activation at the onset of locomotor activity while the activity decreased during continuous movements. Interestingly, the movement-onset related signals were more prominent in the dorsal, compared to the ventral, striatum, whereas the ventral striatum contained more reliable signals for reward-related events than the dorsal striatum. In addition, the authors compared the simultaneously recorded activity of ChIs and dopaminergic inputs using bilateral fiber photometry from the dorsal striatum. The results indicated that the activity of ChIs is correlated with a rapid change in movement (movement state transitions) regardless of whether it is associated with continuous movement. By contrast, dopamine transients occurred predominantly at transitions to locomotion while rapidly-terminated short movements ('jerks') were associated with a decrease in activity. From these results, the authors propose a new model in which ChIs marks movement state transitions while dopamine signals the drive to enact or invigorate a triggered movement.

The reviewers thought that this study provides the most comprehensive and interesting assessment to date of cholinergic interneuron activity and its relationship to dopamine signaling and movement. All the reviewers thought that this study provides novel and important insights into the role of ChIs in movement controls. The manuscript is comprehensive and well written. There was, however, some disagreement among the reviewers as to the overall impact of the work. For instance, one reviewer thought that the conclusion is weak because the literature points to more roles of ChIs in learning than in movements, and the temporal resolution of the calcium indictor (GCaMP) limits mechanistic conclusions about the how these signals come about and how they interact. During discussion, it was pointed out that the authors discuss previous literature on the role of ChIs in movement, and the author's observation of tri-phasic signals in response to salient events which are consistent with single-unit recording provides some assurance of the resolution of the present method. Other reviewers also raised some interpretational concerns that need to be addressed. We would like to see the authors’ response to these issues before publication of this work at *eLife*.

Essential revisions:

1) Detailed histology is needed. We would like to see histology to know where the fibers or cannulae implanted in the dorsal and ventral recording sites were (those are huge targets!). In addition, while we appreciate the example images demonstrating overlap between ChAT immunofluorescence and GCaMP fluorescence in dorsal and ventral striatum, example images are not sufficient – the specificity (and penetrance) of GCaMP expression for ChAT+ neurons should be quantified within each area in multiple brain slices/animals.

2) The last sentence of the Abstract states that transient cholinergic signaling without dopamine input may lead to bradykinetic movements. This is an interesting and easily testable hypothesis. A few sentences later in the Introduction, the authors mention that bilateral optogenetic stimulation of ChIs reduces spontaneous locomotion, which seems quite at odds with the statement from the Abstract. It appears odd that the authors finish the Abstract with a statement that seems to be contradicted several sentences later by previous data.

3) Given the previous literature on the cholinergic pause and its potential role in attention, and the remarkable pause observed during "resets" which could correspond to periods in which the animal may be pausing to assess some kind of internal or external state change, it would be useful to have a bit more discussion about pauses.

4) Given the authors previous work (Howe and Dombeck, 2016) in which optogenetic activation of dopamine neurons leads to short latency onset of locomotion (within 100-200ms), it was intriguing to see here that dopamine transients occurring with low ACh did not elicit locomotion (Figure 6L). It would be useful for the authors to at least comment on this, particularly with regard to their previous data (e.g. did the authors find previously that optogenetic activation of dopamine axons only led to locomotion on some trials?).

5) One reviewer provided a more critical overall evaluation on this work than the other reviewers: "The temporal resolution of GCaMP6f, as used here, is too limited to draw clear mechanistic conclusions about how these signals come about and how they interact. The lack of any behavioral task makes it very hard to infer the functional significance of the signals they observe. The conceptual model they propose is not very conceptual, but more like a descriptive re-statement of the correlations they observed. For these reasons I did not feel that the work provides a substantial advance in our understanding of this circuitry." Please provide your response to these criticisms.

---

## [Author Response]

Essential revisions:1) Detailed histology is needed. We would like to see histology to know where the fibers or cannulae implanted in the dorsal and ventral recording sites were (those are huge targets!). In addition, while we appreciate the example images demonstrating overlap between ChAT immunofluorescence and GCaMP fluorescence in dorsal and ventral striatum, example images are not sufficient – the specificity (and penetrance) of GCaMP expression for ChAT+ neurons should be quantified within each area in multiple brain slices/animals.

In response to the reviewers’ concern regarding the specificity of GCaMP labeling, we have included new histological images and quantification (Figure 1—figure supplement 1A-D) to confirm that expression of GCaMP6f is indeed restricted to ChAT+ cholinergic interneurons in dorsal and ventral striatum. Expression was widespread and only a very small number of false positives were found across three animals and multiple slices quantified (Figure 1—figure supplement 1, mean 2.7% false positives in dorsal and 1.7% in ventral). This small proportion is likely to be an overestimate as strict criteria were applied to identify ChAT/GCaMP+ cells. We have also added a representative coronal brain section showing the track of the optical fiber (Figure 1—figure supplement 1B and additional schematic sections highlighting the range of cholinergic interneuron and dopamine terminal recording locations within the striatum (all photometry recordings were performed within the highlighted regions; Figure 1—figure supplement 1A). The cannulae location and size used here was the same as in Howe and Dombeck, 2016 (Extended Data Figure 2I).

2) The last sentence of the Abstract states that transient cholinergic signaling without dopamine input may lead to bradykinetic movements. This is an interesting and easily testable hypothesis. A few sentences later in the Introduction, the authors mention that bilateral optogenetic stimulation of ChIs reduces spontaneous locomotion, which seems quite at odds with the statement from the Abstract. It appears odd that the authors finish the Abstract with a statement that seems to be contradicted several sentences later by previous data.

We agree with the reviewers that our hypothesis that striatal cholinergic signals, without intact dopamine signaling, may contribute to bradykinetic movements in PD seems to contradict studies that have reported decreases in locomotion with ChI optogenetic stimulation (Kondabolu et al., 2016). Synchronous ChI population transients were most prominent around the onsets of movement and at locomotion ‘resets’, regardless of whether locomotion was successfully initiated and maintained (Figure 1G, H and Figure 6I, J). ChI transients were highly predictive of rest to movement transitions (more so than DA, Figure 6—figure supplement 1A, B). Synchrony and large population transients in ChIs were diminished during transitions to stable, continuous locomotion, as DA signals increased (Figures 1 and 4H). Based on these findings, we hypothesize that synchronous ChI transients may drive changes in movement state, while DA signaling and the accompanying desynchronization of ChI populations serves to enact or maintain those states (i.e. locomotion, Figure 7—figure supplement 1). Artificial synchronous stimulation of ChIs would thus be predicted to generate a state of constant ‘state’ transitions which would prohibit sustained locomotion and perhaps lead to slow, jerky movements from rest. Alternatively, ChI signals at movement onsets may compete with pro-kinetic movement signals (though this possibility is not well supported by our data, Figure 6). Previous studies were limited to coarse assessments of locomotion (Kondabolu et al., 2016) so may not have been sensitive to rapid state dependent alterations in movement patterns, such as the jerks described here. Future manipulation experiments with careful behavioral analyses will be needed to conclusively test these alternatives. Because of the detail required to explain our hypothesis, and the differences compared to the previous optogenetic studies, we have deleted the statement in the Abstract and instead present our hypotheses (with added clarification) in the Discussion section (subsection “Coordinated and divergent cholinergic and dopaminergic movement signaling in dorsal striatum”).

3) Given the previous literature on the cholinergic pause and its potential role in attention, and the remarkable pause observed during "resets" which could correspond to periods in which the animal may be pausing to assess some kind of internal or external state change, it would be useful to have a bit more discussion about pauses.

We have added additional discussion and references about pauses to the Discussion section (subsection “Transient cholinergic interneuron population synchrony marks movement state transitions”). However, the pause cited by the reviewers at resets (we assumed they are referring to Figure 1J) may partly be a result of our selection criteria for significant transient onset alignment (the signaling by definition must be near baseline prior to a transient onset); though we agree that there does appear to be a slight dip just prior to the transient increases in some cases (see example in Figure 1B and Figure 1—figure supplement 1F). Moreover, the reviewers’ idea that brief ChI pauses (followed by rebound bursts) during ongoing behavior may trigger internal behavioral resets, similarly to how (pause/burst) responses to external sensory stimuli may elicit behavioral state changes, is an interesting possibility.

4) Given the authors previous work (Howe and Dombeck, 2016) in which optogenetic activation of dopamine neurons leads to short latency onset of locomotion (within 100-200ms), it was intriguing to see here that dopamine transients occurring with low ACh did not elicit locomotion (Figure 6L). It would be useful for the authors to at least comment on this, particularly with regard to their previous data (e.g. did the authors find previously that optogenetic activation of dopamine axons only led to locomotion on some trials?).

We appreciate the opportunity to address this important point. Our previous work (Howe and Dombeck, 2016) found that stimulation of dopaminergic terminals in the dorsal striatum could rapidly (~200ms) increase the probability of mice initiating locomotion from rest. However, while this was significant on average and often on the individual stimulations within a session, the effectiveness of the stimulation to promote locomotion varied significantly across mice, sessions, and trials (i.e. not every stimulation led to locomotion initiation, see for example the lack of locomotion initiation in a few of the stimulations in Figure 3D of Howe and Dombeck, 2016, and locomotion was not induced with stimulation in every session, see for example the lack of locomotion initiation in some sessions in Figure 3E of Howe and Dombeck, 2016). Moreover, the stimulations were conducted in mice which had been pre-exposed to the treadmill and exhibited regular transitions between locomotion and rest (Howe and Dombeck, 2016, Methods). Thus, it seems likely given our current findings on the involvement of Chls, that dopamine terminal stimulations were most effective in promoting transitions to locomotion when they happened to coincide with spontaneous elevations in ChI signaling (and likely other cell types). By this view, our stimulation of dopamine terminals actually served to boost drive or enact a spontaneous change in movement, rather than triggering the movement per se. We have added discussion in the main text to clarify this important issue (subsection “Coordinated and divergent cholinergic and dopaminergic movement signaling in dorsal striatum”, second paragraph).

5) One reviewer provided a more critical overall evaluation on this work than the other reviewers: "The temporal resolution of GCaMP6f, as used here, is too limited to draw clear mechanistic conclusions about how these signals come about and how they interact.”

We presume that the reviewer is pointing out that we cannot say conclusively with calcium imaging whether there is a direct interaction between the measured DA terminal and ChI signals which generates the behavioral dynamics we describe. Indeed we are careful not to make any firm claims about direct interactions or cellular mechanisms. We agree that GCaMP6f signaling alone cannot establish mechanistic causality in the same way that electrophysiological work in vitro can. However, the point of the study and the value of the calcium imaging approaches we have utilized is to establish the natural fluctuations of DA and ChI signaling on the <~100s of ms timescale during behavior, which is currently unknown and may contribute to the ongoing regulation of movement. This is an essential step towards establishing the dynamic roles of these neurotransmitter systems and constrains the range of mechanisms (described in vitro) which may be responsible for generating these dynamics. The <~100ms temporal resolution provided by GCaMP6f (see Chen et al., 2013) is more than adequate to provide insight into the co-regulation of DA terminal and ChI activity in dorsal striatum. For example, we find that ChI signals at movement initiations are followed (within <100ms) by elevations in DA terminal signaling at locomotion onsets. This is consistent with the possibility of synchronous ChI signals positively modulating DA terminal activity (as described in vitro e.g. Threlfell et al., 2012). However, this modulation does not appear to be deterministic (i.e. every synchronous ChI event leads to elevated DA signaling) during behavior because ChI signals at the onsets of rapidly terminated jerks are associated with decreases in DA terminal signaling (Figure 6J), indicating partially independent control of ChI and DA terminal signaling. This was unknown before our GCaMP6f study. Of course, future work, possibly using targeted manipulations and neuromodulator specific sensors to directly measure release (e.g. Patriarchi et al., 2018), will be necessary to establish the precise mechanisms responsible for the dynamics we observe.

“The lack of any behavioral task makes it very hard to infer the functional significance of the signals they observe.”

We chose to study ChI and DA terminal signaling during spontaneous movement because both of these neurotransmitters have been shown in animals and in humans with clinical disorders to be involved in the non-task dependent modulation of movement drive. This involvement is not explicitly dependent on instrumental aspects of goal-directed behavior or on Pavlovian cues (the most common types of task historically used to study these systems). Humans with dysfunction of dorsal striatum neuromodulator signaling (such as in Parkinson’s Disease) exhibit general (non task-specific) movement deficits. Experimental manipulations (lesions, pharmacology, optogenetics) of DA and ChI signaling in animal models likewise alter basic aspects of movement drive such as the probability of initiation and ongoing movement vigor. Finally, a number of recent in vivo recording studies have also provided support for the role of dorsal striatum ChI and DA signaling in the ongoing control of (internally generated) spontaneous movement (e.g. Howe and Dombeck, 2016, Dodson et al., 2016, DaSilva et al., 2018). Our work has addressed the critical, unresolved question of how naturally occurring neuromodulator dynamics in the striatum contribute to general aspects of ongoing movement drive. We avoided an instrumental or Pavlovian task design to isolate signaling related to the initiation and maintenance of spontaneous movement from potentially confounding variables such as sensory cues or internally generated reward predictions. However, we note that we did employ a behavioral task (classical conditioning paradigm) to confirm the existence of previously described signaling in ChIs related to reward predictive cues (Figure 2—figure supplement 1). Further studies will be required to determine how the signals we have described are modulated during goal-directed instrumental tasks.

“The conceptual model they propose is not very conceptual, but more like a descriptive re-statement of the correlations they observed.”

We included the schematics in Figure 7—figure supplement 1 to visually summarize the results and hypotheses outlined in the Discussion. We agree that it is more a summary than a ‘model’ so have corrected the wording in the text and legend accordingly.